# Separation and Bias of Deep Equilibrium Models on Expressivity and Learning Dynamics

**Zhoutong Wu**[1]
ztwu@stu.pku.edu.cn

**Yimu Zhang**[2]
zym24@stu.pku.edu.cn

**Cong Fang**[2,3][†]
fangcong@pku.edu.cn

**Zhouchen Lin**[2,3,4][†]
zlin@pku.edu.cn

[1] Academy for Advanced Interdisciplinary Studies, Peking University
[2] State Key Lab of General AI, School of Intelligence Science and Technology, Peking University
[3] Institute for Artificial Intelligence, Peking University
[4] Pazhou Laboratory (Huangpu), Guangzhou, Guangdong, China

## Abstract

The deep equilibrium model (DEQ) generalizes the conventional feedforward neural network by fixing the same weights for each layer block and extending the number of layers to infinity. This novel model directly finds the fixed points of such a forward process as features for prediction. Despite empirical evidence showcasing its efficacy compared to feedforward neural networks, a theoretical understanding for its separation and bias is still limited. In this paper, we take a step by proposing some separations and studying the bias of DEQ in its expressive power and learning dynamics. The results include: (1) A general separation is proposed, showing the existence of a width-$m$ DEQ that any fully connected neural networks (FNNs) with depth $O(m^\alpha)$ for $\alpha \in (0, 1)$ cannot approximate unless its width is sub-exponential in $m$; (2) DEQ with polynomially bounded size and magnitude can efficiently approximate certain steep functions (which has very large derivatives) in $L^\infty$ norm, whereas FNN with bounded depth and exponentially bounded width cannot unless its weights magnitudes are exponentially large; (3) The implicit regularization caused by gradient flow from a diagonal linear DEQ is characterized, with specific examples showing the benefits brought by such regularization. From the overall study, a high-level conjecture from our analysis and empirical validations is that DEQ has potential advantages in learning certain high-frequency components.

## 1 Introduction

Implicit deep learning [1], a paradigm that generalizes the recursive principles of traditional explicit models, has gained renewed interest with the advent of novel neural network architectures. Among these, deep equilibrium model (DEQ) [2] stands out as a commonly utilized model. In contrast to explicit neural network that derives features through forward propagation, DEQ computes features directly by solving an equilibrium equation induced by the implicit layer. Since the equilibrium state is also the limit point of the infinitely recursive iterations of the implicit layer, DEQ can be regarded as a new neural network that models the limit of a multi-layer weight-tied neural network with the depth going to infinity.

---

[†]Corresponding author.

38th Conference on Neural Information Processing Systems (NeurIPS 2024).

Nowadays, DEQ has become a popular and widely studied model in the field of machine learning. On the empirical side, competitive performances against explicit feedforward neural networks have been achieved in various real applications such as natural language processing [2], computer vision [3], image generation [4], and solving inverse problems [5]. On the theoretic side, a main research line is to study the well-posedness of DEQ. This line aims to analyze when unique equilibrium can be guaranteed by DEQ and some weight parameterization and initialization techniques have been proposed to ensure the well-posedness [6, 7, 8].

However, despite wide studies on DEQ, an understanding of the basic learning theory for its separation and bias against explicit feedforward neural networks is still limited. For the expressivity, a preliminary study about the connections between DEQ and fully-connected network (FNN) is provided in the seminar work [2], where it is proved that every FNN can be reformulated as a large DEQ under a specific weight re-parameterization, whereas, a deeper study on the provable and quantitative advantage of DEQ in its expression power is still lacking. Besides, there is another research line that studies the learning properties of DEQ using the so-called neural tangent kernel (NTK) view [9], originating from analyzing FNNs [10, 11]. It has been shown that under suitable initialization, the dynamic of over-parameterized DEQ can be approximated by a linear kernel model [12, 13], therefore the global convergence of gradient descent algorithm and possible generalization can be achieved under some regimes. However, in the NTK regime, it is shown that DEQs are almost equivalent to not-so-deep explicit FNNs for high dimensional Guassian mixtures [14], and it is still not known whether DEQs have potential advantages over FNNs in more general settings. A study on the separation and bias of DEQ over FNN can provide us with clear and intuitive suggestions about when DEQ is preferred in practice, thus it is strongly desired. In this paper, we initialize the study by analyzing the expressive power and learning dynamics of DEQs. The main results are sketched as follows.

1. We first propose a general separation showing that there exists a width-$m$ DEQ which cannot be approximated to a constant accuracy by an FNN with depth $O(m^\alpha)$ for $\alpha \in (0, 1)$ unless its width is $\exp(\Omega(m^{1-\alpha}))$. This is achieved by comparing the the number of linear regions that the two networks can generate. Based on the result, we further prove that a width-$m$ DEQ can generate at most $2^m$ linear regions, which has provable advantages than FNNs.

2. We then propose another separation, where a steep function in $[0, 1]^d$ being the solution to fixed point equation is considered as the target function. We show that a DEQ with size and magnitude bounded by $O(\varepsilon^{-1})$ can approximate this function to $O(\varepsilon)$-accuracy in $L^\infty$ norm, whereas an FNN with bounded depth and exponentially bounded width cannot unless its weights is $\exp(\Omega(d))$. For the technical contribution, we manage to show that an approximation of the fixed point mapping by the implicit layer can also guarantee the approximation the solution defined by the fixed point equation even if the Lipschitz constant of the fixed point mapping is very close to 1 by a new observation as shown in Lemma 3.

3. Finally, we study the bias of DEQ from the perspective of learning dynamics. We propose a general characterization of regularization for gradient flow in an overparameterized setting. We further analyze the dynamics of both gradient flow and gradient descent, showing that under mild conditions, convergence is guaranteed, and the model tends to produce 'dense' features. Then we offer a concrete example on a specific Out-of-Distribution (OOD) task, demonstrating that this bias can help reduce the OOD error.

Finally, we conduct experiments to validate our theoretical results. From the overall study, a high-level conjecture is that DEQ has potential advantages in learning certain high-frequency components.

**Notations.** We use standard notation $O(\cdot)$ and $\Omega(\cdot)$ to hide constants. We use $\sigma$ to denote the ReLU function, i.e., $\sigma(x) = \max(0, x)$, and we use $\text{sgn}(\cdot)$ to denote the sign function. We use $\text{diag}(\cdot)$ to transform a vector into a diagonal matrix with the vector's elements on the diagonal. We denote by $\| \cdot \|_p$ the $\ell^p$ vector norm or the subordinate matrix norm, and by $\|h\|_{L^p(k)}$ the $L^p$-norm of a function $h$ on a compact set $K$. For a vector or vector-valued function $\mathbf{v}$, we denote $v_i$ the $i$-th entry of the vector or the function. For a function $u : \mathbb{R} \to \mathbb{R}$, we denote $u^{\circ n}$ the $n$-fold composition of $u$.

## 2  Related Works

In this section we briefly review the literature that are most related to us.

**Theoretical Studies on DEQs.** Theoretical research on DEQs has primarily focused on ensuring their well-posedness [6, 7]. To guarantee well-posedness, strategies like new parameterizations [6, 7], regularization [15], and special initialization [8] were proposed. Another research line delves into the learning properties of DEQ. The expressivity of DEQ is preliminarily studied in [2]. Regarding learning dynamics, some works [16, 17, 13] couple the dynamics of over-parameterized DEQs with a linear kernel using the NTK method. They manage to prove the global convergence and study the generalization [17]. Other studies examine DEQs in a linear framework or infinite-width limit. For instance, Kawaguchi [16] studies the convergence of linear DEQ, while Gao et al. [18] explore information propagation in DEQ and show its equivalence to a Gaussian process in the infinite-width limit. Nevertheless, an in-depth study on the potential or quantifiable advantage of DEQ over FNN is still lacking.

**Separations on Expressivity of Neural Networks.** The separation on expressivity of neural networks is a fundamental study characterizing functions that can be approximated efficiently by one neural architecture but cannot by another. These architectures include FNNs [19, 20], CNNs [21], RNNs [22], etc. Since DEQ can be viewed as an infinitely deep weight-tied network, depth separation [23] is most relevant to our study. A key study by Telgarsky [24] constructs a saw-tooth function with many oscillations to give a separation, which further inspires a series of separations [25, 26, 27]. In addition to depth, some recent works study the separation regrading the overall number of neurons [28] or the magnitude of parameters [29] of the networks. In this paper, the first separation is also inspired by Telgarsky's construction, whereas we focus on the separation between DEQ and FNN and provide a more refined analysis of networks' depth. The second separation is new.

**Implicit Bias of Learning Dynamics on Neural Networks.** The implicit bias of learning dynamics plays a key role in determining what particular optima can be found by the algorithms when there are multiple optima. A series of papers study the implicit regularization of gradient-based methods, showing that under varying settings, these algorithms bias towards solutions with specific properties [30, 31], such as norm minimization [32], sparsity [33] and low complexity [34, 35, 36]. Due to the theoretical barrier in analyzing nonlinear neural networks [37], most existing works focus on simplified models such as random feature models [38, 32], networks with quadratic activations [39] and diagonal linear networks [33]. This paper follows similar strategies and analyzes the implicit bias of a simplified diagonal linear DEQ from learning dynamics.

## 3  Preliminaries of DEQ

The DEQ is an implicit-depth model [2] that employs the same weights in each layer block of a feedforward neural network and extends the number of layer to infinity. The layer blocks used in DEQ can be fully connected, convolutional, or Transformer blocks, resulting in different variants of deep equilibrium networks. In this paper, we consider a vanilla DEQ with ReLU activation as the generalization of an FNN for the separation result and a simplified linear diagonal DEQ for the bias analysis. Specifically, an $L$-layer FNN from $\mathbb{R}^d$ to $\mathbb{R}^s$ can be expressed as

$$\mathbf{z}^1 = \mathbf{x}; \quad \mathbf{z}^{i+1} = \sigma(\mathbf{W}_i \mathbf{z}^i + \mathbf{b}_i), \quad 1 \le i \le L-2; \quad \mathbf{y} = \mathbf{W}_L \mathbf{z}^L, \tag{1}$$

where $\mathbf{x} \in \mathbb{R}^d$ and $\mathbf{y} \in \mathbb{R}^s$. In DEQ, each $\mathbf{W}_i$ and $\mathbf{b}_i$ in Eq. (1) is replaced by the same weight $\mathbf{W}$ and bias $\mathbf{b}$, and a linear transform of the input $\mathbf{U}\mathbf{x}$ is added to each layer, i.e., $\mathbf{z}^l = \sigma(\mathbf{W}\mathbf{z}^{l-1} + \mathbf{U}\mathbf{x} + \mathbf{b})$ for all $l$. By extending the layer $l$ to infinity, the feature and the prediction of this DEQ can be expressed as

$$\begin{aligned} \mathbf{z} &= \sigma(\mathbf{W}\mathbf{z} + \mathbf{U}\mathbf{x} + \mathbf{b}), \\ \mathbf{y} &= \mathbf{A}\mathbf{z}, \end{aligned} \tag{2}$$

where $\mathbf{W} \in \mathbb{R}^{m \times m}, \mathbf{U} \in \mathbb{R}^{m \times d}, b \in \mathbb{R}^m$, and $\mathbf{A} \in \mathbb{R}^{s \times m}$. We call $\sigma(\mathbf{W}\mathbf{z} + \mathbf{U}\mathbf{x} + \mathbf{b})$ the implicit layer and $m$ the width of DEQ. In this paper, we mainly consider $s = 1$, i.e., DEQ as a scalar function on $\mathbb{R}^d$.

In [2], the authors show that every FNN can be reformulated as a large DEQ with specific weight reparameterization. Specifically, the depth-$L$ FNN described in Eq. (1) is equivalent to a DEQ in the

form of Eq.(2) with

$$\mathbf{A} = \begin{pmatrix} \mathbf{0}, & \cdots & , \mathbf{W}_L \end{pmatrix}, \mathbf{W} = \begin{pmatrix} \mathbf{0} & & & & \\ \mathbf{W}_2 & \mathbf{0} & & & \\ & \mathbf{W}_3 & \mathbf{0} & & \\ & & \ddots & \ddots & \\ & & & \mathbf{W}_{L-1} & \mathbf{0} \end{pmatrix}, \mathbf{U} = \begin{pmatrix} \mathbf{W}_1 \\ \mathbf{0} \\ \vdots \\ \mathbf{0} \end{pmatrix}, \mathbf{b} = \begin{pmatrix} \mathbf{b}_1 \\ \mathbf{b}_2 \\ \vdots \\ \mathbf{b}_{L-1} \end{pmatrix}. \quad (3)$$

## 4   Separation on the Expressivity of DEQ

In this section, we focus on the separations on the expressivity of DEQ with ReLU activation. We follow the common ways to compare the expressivity from the actual size (width and depth) of the networks. More explanations on the fairness of the comparison are provided in Appendix A.4. We will show that DEQ is more *parameter-efficient* in approximating specific target functions than FNN.

### 4.1   General Separation over FNNs

The following theorem states a general separation between DEQ and FNN from the size of networks. The motivation behind the theorem is a common observation that functions with many linear pieces are typically hard to be approximated by functions having fewer linear pieces.

**Theorem 1.** *Let $m \in \mathbb{N}^+$. Assume that $L \leq m^\alpha$ for some $0 < \alpha < 1$. Then there exists a function $N_d : [0,1]^d \to \mathbb{R}$ computed by a width-$m$ ReLU-DEQ, such that for any function $N_f : [0,1]^d \to \mathbb{R}$ computed by a depth-$L$ ReLU- FNN with width at most $2^{m^{1-\alpha}-1}$, it holds that*

$$\int_{[0,1]^d} |N_d(\mathbf{x}) - N_f(\mathbf{x})| \mathrm{d}\, \mathbf{x} \geq \frac{1}{16}.$$

The proof is provided in Appendix A.1. It involves quantifying the number of linear regions[1] generated by a DEQ compared to an FNN. Specifically, we show in the proof that there exists a DEQ producing $2^m$ linear pieces whereas no-so-deep FNNs, i.e., FNNs with depth $O(m^\alpha)$ cannot generate such a large number of linear regions unless the width is sub-exponentially large.

Moreover, the example of the hard-to-approximate DEQ enables us to derive an exact bound on the number of linear regions that a DEQ can generate. This result is of independent interest and is stated in the proposition below.

**Proposition 1.** *Let $m > 0$. A width-$m$ ReLU-DEQ has at most $2^m$ linear regions in the input space. Moreover, this upper bound is attainable, i.e., there exists a width-$m$ ReLU-DEQ that computes a function with $2^m$ linear regions on $\mathbb{R}^d$.*

**Remark 1.** *As a comparison, the work of [40] analyzes ReLU-FNNs. It shows that for a ReLU-FNN with a total of $\tilde{N}$ neurons of arbitrary depth, the maximal number of linear regions is bounded above by $2^{\tilde{N}}$. To the best of our knowledge, it is yet to be determined whether this bound is achievable. Moreover, there is evidence suggesting that this upper bound is not achievable for FNNs that when the input dimension is $1$ (See Lemma 5 in Appendix A.1 for details). Consequently, width-$m$ DEQs can potentially generate a larger number of linear regions compared to FNNs with $m$ neurons, as DEQs have been shown to achieve their upper bound.*

Theorem 1 shows that there exists a width-$m$ DEQ that is hard to be approximated by FNN with depth $O(m^\alpha)$. This theorem along with Proposition 1 reveals that, although DEQ computes features by solving an equilibrium function induced by a shallow implicit layer, its complexity in terms of expressing linear regions of DEQ can be larger than that of not-so-deep FNN.

### 4.2   Separation on Certain Steep Functions

In this section, we present another separation concerning both the size and parameter magnitude of neural networks, which more explicitly reveals the bias and potential advantages of DEQ on

---

[1]We follow the definition of linear regions in [40]: For any piecewise linear function $F : \mathbb{R}^{n_0} \to \mathbb{R}$, a linear region of the function is a connected subset $D \subset \mathbb{R}^{n_0}$ satisfying 1) $F$ is linear on $D$; 2) If $F$ is linear on some connected set $\tilde{D} \supset D$, then $\tilde{D} = D$.

expressivity. The separation is based on the observation that the fixed point of a DEQ can be rewritten as the solution to an optimization problem under certain conditions.

To be specific, consider a simple quadratic optimization problem with the optimization variable $\mathbf{z} \in \mathbb{R}^m$ and a parameter $\mathbf{x} \in \mathbb{R}^d$:

$$\min_{\mathbf{z}} \ \frac{1}{2}\mathbf{z}^T\mathbf{A}(\mathbf{x})\,\mathbf{z}+\mathbf{b}^T(\mathbf{x})\,\mathbf{z}+\mathbf{c}, \tag{4}$$

where $\mathbf{A}(\mathbf{x})$ is a positive definite matrix parameterized by $\mathbf{x}$ and $\eta\mathbf{I} \succ \mathbf{A}(\mathbf{x}) \succ \mathbf{0}$ for some $\eta > 0$. Approximating $\mathbf{z} = \mathbf{z}(\mathbf{x})$, i.e., the optimum as a function of the parameter $\mathbf{x}$, serves useful primitives in various applications. Directly approximating $\mathbf{z}(\mathbf{x})$ by FNN requires the approximation of $\mathbf{z}(\mathbf{x}) = -\mathbf{A}(\mathbf{x})^{-1}\mathbf{b}(\mathbf{x})$. On the other hand, from the optimality condition, $\mathbf{z}(\mathbf{x})$ is implicitly defined through fixed point equation

$$\mathbf{z} = \mathbf{z} - \frac{1}{\eta}\left(\mathbf{A}(\mathbf{x})\,\mathbf{z}+\mathbf{b}(\mathbf{x})\right).$$

Hence, approximating $\mathbf{z}(\mathbf{x})$ by DEQ may only require the approximation of the fixed point mapping $\mathbf{z} - \frac{1}{\eta}\left(\mathbf{A}(\mathbf{x})\,\mathbf{z}+\mathbf{b}(\mathbf{x})\right)$ by the implicit layer. To some extent, the approximation problem is 'altered' due to the model difference, which possibly leads to distinctive division in approximation.

Now, we construct a workable instance. The objective function of our central interest is a special case of Eq.(4) given by:

$$\min_z (1 + \delta - x_1)z^2 - \delta x_1 z, \quad \mathbf{x} \in [0, 1]^d, \tag{5}$$

where $\delta = 2^{-d}$. The solution function is calculated as

$$g(\mathbf{x}) = \frac{\delta x_1}{2(1 + \delta - x_1)}, \quad \mathbf{x} \in [0, 1]^d, \tag{6}$$

and it can also be determined by the following fixed point equation

$$z = \tilde{g}(z, \mathbf{x}) := (x_1 - \delta)z + \frac{1}{2}\delta x_1. \tag{7}$$

Note that $g(\mathbf{x})$ has very large derivative when $x_1$ is near 1. It can be regarded as a continuous version of the common indicator function of the first entry $\frac{1}{2}\mathbf{1}_{x_1=1}(\mathbf{x})$. The separation is presented as follows.

**Theorem 2.** *Let $g(\mathbf{x})$ be defined as in Eq.(6) for $\mathbf{x} \in [0, 1]^d$ and $\frac{1}{4} \geq \varepsilon > 0$.*

- A. *For any function $N_{fnn}(x)$ implemented by an FNN with depth $L$ and width $k$ where $L \leq C$ and $k \leq 2^{\frac{d}{2C}}$ for some constant $C = O(1)$. If*

$$\|N_{fnn}(\mathbf{x}) - g(\mathbf{x})\|_{L^\infty([0,1]^d)} \leq \frac{1}{16},$$

  *then there exists a weight parameter $W_{ij}$ of the FNN for $1 \leq i \leq L$ and $1 \leq j \leq k$, such that*

$$|W_{ij}| \geq 2^{\frac{d}{2C}}.$$

- B. *There exists a function $N_{deq}$ implemented by a DEQ with width bounded by $5\varepsilon^{-1}$ and weights bounded $2\varepsilon^{-1}$, such that*

$$\|N_{deq}(\mathbf{x}) - g(\mathbf{x})\|_{L^\infty([0,1]^d)} \leq \varepsilon.$$

**Remark 2.** *The inapproximability result of FNN in Theorem 2 is stated from the perspective of weight magnitude, which holds practical significance. Exponentially large weight often results in exponential iterations of optimization algorithms in learning with this model, as also noted in [41]. Additionally, neural networks in practice typically have small weights due to techniques such as (standard) small initialization, normalization, and gradient clipping.*

The proof is shown in Appendix A.2. In Theorem 2, the inapproximability of FNNs is relatively simple: Direct calculation shows that the derivative of the target function $g(\mathbf{x})$ is exponentially large when $x_1 > 1 - \delta$. To approximate $g(\mathbf{x})$ in $L^\infty$ norm requires FNNs to have large derivative in certain region, resulting in exponentially large weight for FNNs with bounded depth. On the other hand, the

proof of the approximability of DEQs is more technical. While $\tilde{g}$ in Eq. (7) seems more benign, it is not clear how to construct the approximation using the implicit layer in Eq. (2) that resembles an 1-layer FNN with very limited expressive power. Moreover, even if we manage to approximate $\tilde{g}$ in Eq. (7), it will not necessarily imply a good approximation between the fixed point of DEQ and the solution of $z = \tilde{g}(z, \mathbf{x})$, i.e., the target function due to the Lipschitz constant of $\tilde{g}$ with respect to $z$ being very close to 1 when $x_1$ is around 1 according to Eq. (7). We provide a proof sketch of this result in Section 4.3.

Further insights and implications can be gleaned from Theorem 2. First, it suggests that DEQ may excel in approximating functions induced by fixed-point iterations. In other words, DEQ may be better suited for representing algorithms. Second, Theorem 2 implies that functions with large derivative, or high-frequency components, may be approximated more efficiently by DEQ, as the function to be approximated by the implicit layer can have much smaller derivative.

### 4.3 Proof Sketch of B. in Theorem 2

As discussed in Section 4.2, we want to approximate $\tilde{g}$ using the implicit layer of DEQ. Due to the limited expressive power of the implicit layer, we propose an equivalent reparameterization of DEQ.

**Lemma 1.** *Consider a revised DEQ defined as*

$$\begin{aligned}
\mathbf{z} &= \mathbf{V}\sigma(\mathbf{W}\,\mathbf{z} + \mathbf{U}\,\mathbf{x} + \mathbf{b}), \\
\mathbf{y} &= \mathbf{B}\,\mathbf{z},
\end{aligned} \tag{8}$$

*where $\mathbf{x} \in \mathbb{R}^d, \mathbf{z} \in \mathbb{R}^m, \mathbf{W} \in \mathbb{R}^{q \times m}, \mathbf{U} \in \mathbb{R}^{q \times d}, \mathbf{V} \in \mathbb{R}^{m \times q}, b \in \mathbb{R}^q, \mathbf{B} \in \mathbb{R}^{p \times m}$ and $\|\mathbf{W}\,\mathbf{V}\|_2 \leq 1$. Then any revised DEQ can be represented by a vanilla DEQ defined as in Eq. (2) with width $q$.*

Lemma 1 enables us to approximate $\tilde{g}(z, \mathbf{x})$ using the revised implicit layer, denoted by $\tilde{h}(z, \mathbf{x})$. Then the crux of the proof centered in bounding the error between the equilibria of two fixed-point equations. To begin, for every $\mathbf{x}$ we denote $\hat{u}(z) = z - \tilde{g}(z, \mathbf{x})$, $\hat{v}(z) = z - \tilde{h}(z, \mathbf{x})$ and consider $|\hat{u}^{\circ 2}(z) - \hat{v}^{\circ 2}(z)|$. Suppose that $\hat{u}(z)$ is $L_{\hat{u}}$-Lipschitz, then we have

$$|\hat{u}^{\circ 2}(z) - \hat{v}^{\circ 2}(z)| \leq |\hat{u}^{\circ 2}(z) - \hat{u} \circ \hat{v}(z)| + |\hat{u} \circ \hat{v}(z) - \hat{v}^{\circ 2}(z)| \leq (L_{\hat{u}} + 1)|\hat{u}(z) - \hat{v}(z)|.$$

Thus if $L_{\hat{u}} < 1$, by recursion, we can bound distance the between the infinitely composition of $\hat{u}(z)$ and $\hat{v}(z)$, from which the error of the two fixed points can be bounded.

**Lemma 2.** *Let $\Omega \subset \mathbb{R}$ be a compact set, and $u(z, \mathbf{x}), v(z, \mathbf{x}) : \Omega \times [0, 1]^d \to \Omega$ be two functions. Assume that for all $\mathbf{x} \in [0, 1]^d$, $u(\cdot, \mathbf{x})$ and $v(\cdot, \mathbf{x})$ are Lipschitz continuous with Lipschitz constant $L_u, L_v < 1$, respectively. Then for any $\mathbf{x} \in [0, 1]^d$, it holds that*

$$|z_u(\mathbf{x}) - z_v(\mathbf{x})| \leq \min\{(1 - L_u)^{-1}, (1 - L_v)^{-1}\} \cdot \max_{z \in \Omega} |u(z, \mathbf{x}) - v(z, \mathbf{x})|,$$

*where $z_u(\mathbf{x})$ and $z_v(\mathbf{x})$ are the fixed point of $z = u(z, \mathbf{x})$ and $z = v(z, \mathbf{x})$, respectively.*

In our case, $u(z, \mathbf{x})$ and $v(z, \mathbf{x})$ in this Lemma represent $\tilde{g}(z, \mathbf{x})$ and $\tilde{h}(z, \mathbf{x})$, respectively. When $x_1 < 1 - \text{poly}(d)^{-1}$, by calculating $\frac{\partial \tilde{g}(z, \mathbf{x})}{\partial z}$, we have $(1 - L_{\tilde{g}})^{-1} < \text{poly}(d)$. Leveraging this and Lemma 2, we just need $\|\tilde{h} - \tilde{g}\|_\infty \leq \text{poly}(d)^{-1}$ to achieve a final accuracy of $O(\varepsilon)$. However, when $x \geq 1 - \delta$, we only have $(1 - L_{\tilde{g}})^{-1} < \exp(\Omega(d))$, which may necessitate an exponential width for the implicit layer to achieve $O(\varepsilon)$ accuracy. In fact, $\tilde{h}(z, \mathbf{x}) = x_i z$ gives an example that even assuming $\|\tilde{h} - \tilde{g}\|_\infty \leq \exp(\Omega(d))^{-1}$ is not sufficient to achieve $O(\varepsilon)$ accuracy since $z_{\tilde{h}}(1) - z_{\tilde{g}}(1) = \frac{1}{2}$. So it seems difficult to bound the error without a specific structure of $\tilde{h}$. To overcome the issue, we observe a *novel* property that enables us to effectively bound the error.

**Lemma 3.** *Let $\xi > 0$. Under the conditions in Lemma 2, if for any interval $T \subset \Omega$ with $diam(T) > \xi$, $u(z, \mathbf{x}) = v(z, \mathbf{x})$ has a zero in $T$ for all $\mathbf{x}$, then it holds that*

$$|z_u(\mathbf{x}) - z_v(\mathbf{x})| \leq \xi, \quad \forall \mathbf{x} \in [0, 1]^d.$$

The intuition behind Lemma 3 is that if for any $\mathbf{x}$, $z - u(z, \mathbf{x})$ and $z - v(z, \mathbf{x})$ as two monotone univariate functions w.r.t. $z$ can take the same value at frequent intervals, then their zeros will also be close to each other. By using this Lemma, it suffices to construct such $\tilde{h}(z, \mathbf{x})$ that equals $\tilde{g}(z, \mathbf{x})$ at frequent interval of length $O(\varepsilon)$ for every $\mathbf{x}$.

## 5 The Bias on Learning Dynamics of DEQ

In this section, we study the implicit bias of a simplified linear diagonal DEQ and present a concrete example illustrating how such an implicit bias may improve generalization. Specifically, we focus on the overparameterized setting and our analysis is beyond the lazy training regime.

To ensure tractability, we follow the common techniques (e.g. see [31, 39]) to reduce a matrix problem to a vector problem by considering only the diagonal elements of the weight matrix. Our primary focus is on the following model:

$$f(\mathbf{w}, \mathbf{x}) = \sum_{i=1}^{d} \frac{1}{1 - w_i} x_i := \langle \boldsymbol{\beta}, \mathbf{x} \rangle, \quad \beta_i = \frac{1}{1 - w_i}. \tag{9}$$

The model can be regarded as a diagonal linear DEQ in Eq. (2) with activation $\sigma = \text{Id}$, weights $\mathbf{W} = \text{diag}(w_1, w_2, \cdots, w_d)$, $\mathbf{U} = \mathbf{I}_d$, $\mathbf{b} = \mathbf{0}$ and $\mathbf{A} = (1, 1, \cdots, 1)^T \in \mathbb{R}^d$. Although simplified, this model is essentially a nonlinear model and it retains the implicit nature of DEQ.

Our primary focus lies in minimizing the expected square loss:

$$\min_{\mathbf{w}} L(\mathbf{w}) := \frac{1}{2} \mathbb{E}_{(\mathbf{x}, y) \sim \mathcal{D}}[(y - f(\mathbf{w}, \mathbf{x}))^2]. \tag{10}$$

We are given access to a set of i.i.d. training examples $\{(\mathbf{x}_i, y_i)\}_{i=1}^{N}$, and we denote the (half) square loss on these examples by $\hat{L}(\mathbf{w}) = \frac{1}{2} \sum_{i=1}^{N} (y_i - f(\mathbf{w}, x_i))^2$. As mentioned above, we focus on the overparameterized setting that $N \leq d$. Moreover, let

$$\mathbf{X} = (\mathbf{x}_1, \cdots, \mathbf{x}_N)^T, \quad \mu_{\min} = \lambda_{\min}(\mathbf{X}\mathbf{X}^T), \quad \mu_{\max} = \lambda_{\max}(\mathbf{X}\mathbf{X}^T),$$

where $\mu_{\min} > 0$ can hold when $N \leq d$ and the data matrix $\mathbf{X}$ is of full rank. We mainly consider the dynamics of gradient flow (GF) and gradient descent (GD) with fixed stepsize $\eta$ on minimizing $\hat{L}(\mathbf{w})$, expressed as follows

$$\text{(GF)} \quad \dot{\mathbf{w}}(t) = -\nabla_{\mathbf{w}} \hat{L}(\mathbf{w}(t)); \quad \text{(GD)} \quad \mathbf{w}^{k+1} = \mathbf{w}^k - \eta \nabla_{\mathbf{w}} \hat{L}(\mathbf{w}^k). \tag{11}$$

The main theorem below gives a general characterization of the bias of diagonal linear DEQ in the overparameterized regime. The proof is based on the technique proposed in [31].

**Theorem 3.** *Let $\beta_i$ in Eq. (9) be initialed as $\beta_i(0) > 0$ for all $i$. Suppose that gradient flow for the parameterization problem in Eq. (10) converges to some $\hat{\beta}$ satisfying $\mathbf{X}\hat{\beta} = \mathbf{y}$, then*

$$\hat{\boldsymbol{\beta}} = \underset{\boldsymbol{\beta}}{\arg\min} \, Q(\boldsymbol{\beta}), \quad s.t. \, \mathbf{X}\boldsymbol{\beta} = \mathbf{y}, \tag{12}$$

*where $Q(\boldsymbol{\beta}) = \sum_{i=1}^{d} q_i(\beta_i)$ and $q_i(x) = \frac{1}{2x^2} + \beta_i(0)^{-3} x$.*

**Remark 3.** *In Theorem 3, our proof shows that $\boldsymbol{\beta}(t)$ remains positive for all entries throughout the training process. Within the space where $\boldsymbol{\beta} > 0$, $Q(\boldsymbol{\beta})$ is convex and has a unique minimum. The restriction to positive entries arises from our simplification on $\mathbf{A}$ in Eq. (9) to be an all-one vector. To accommodate negative entries, one could assign $-1$ to the corresponding entry. In this case, $q_i(x)$ becomes $q_i(x) = \frac{1}{2x^2} - \beta_i(0)^{-3} x$ with $\beta_i(0) < 0$, which is convex for $x < 0$.*

All the proofs in this section are included in Appendix A.3. The theorem implies that the bias of the (simplified) DEQ significantly differs from that of conventional linear models and two-layer linear network which tends to give a minimum $\ell_2$-norm interpolator [42]. Specifically, the predictor $\hat{\boldsymbol{\beta}}$ hardly admits parameters of small magnitude due to the penalty term $\frac{1}{2} \sum_{i=1}^{d} \frac{1}{\beta_i^2}$. Meanwhile, the predictor can endure parameters of greater magnitude as the penalty $q_i(x)$ increase almost linearly when $x$ is large.

We then study the implicit bias from the learning dynamics of GF and GD. We show that when $\mu_{\min} > 0$, under mild conditions, the convergence of both algorithms is guaranteed. Moreover, in this case, a positive lower bound of the $\ell_\infty$ norm of the iterates can be derived, indicating that the model inclines to produce 'dense' features in learning process.

**Assumption 1.** *Denote by $\boldsymbol{\beta}_0$ the initialization of $\boldsymbol{\beta}$ of the model in Eq. (9). There exists an optima $\hat{\boldsymbol{\beta}}^*$, i.e., $\mathbf{X}\hat{\boldsymbol{\beta}}^* = \mathbf{y}$ and a constant $c > 0$, such that*

$$\|\hat{\boldsymbol{\beta}}^*\|_\infty - \|\hat{\boldsymbol{\beta}}^* - \boldsymbol{\beta}_0\|_2 \geq c > 0.$$

**Theorem 4.** *Let $\{\boldsymbol{\beta}(t)\}$ be the process following GF in Eq. (11) and $\{\boldsymbol{\beta}^k\}$ the iterates following GD in Eq. (11). Assume that $\mu_{min} > 0$ and the initialization $\boldsymbol{\beta}(0)$ and $\boldsymbol{\beta}^0$ satisfy Assumption 1 with an optima $\hat{\boldsymbol{\beta}}^*$.*

   A. *$\{\boldsymbol{\beta}(t)\}$ converges to an optima $\boldsymbol{\beta}_f^\infty$ with $\|\boldsymbol{\beta}_f^\infty\|_\infty \geq c$. Moreover, for any $t \geq 0$, we have*
   $$c \leq \|\boldsymbol{\beta}(t)\|_\infty \leq \|\hat{\boldsymbol{\beta}}^*\|_\infty + \|\hat{\boldsymbol{\beta}}^* - \boldsymbol{\beta}_0\|_2.$$

   B. *If there exists a constant $C > 0$ such that $\|\boldsymbol{\beta}^k\|_\infty \leq C$ for all $k$, then $\{\boldsymbol{\beta}^k\}$ converges to an optima $\boldsymbol{\beta}_d^\infty$ with $\|\boldsymbol{\beta}_d^\infty\|_\infty \geq c$. Moreover, for any $k \geq 0$, we have $c \leq \|\boldsymbol{\beta}^k\|_\infty \leq C$.*

**Remark 4.** *The assumption in Theorem 4 that $\|\boldsymbol{\beta}^k\|_\infty$ is uniformly bounded can be removed if we manually incorporate a constraint on $\boldsymbol{\beta}$ and optimize the problem using projected gradient descent. In practice, certain reparameterization tricks [6, 7] are proposed to ensure that $\mathbf{I} - \mathbf{W} \succeq mI$ for some $m > 0$, thus corresponding to the aforementioned assumption.*

Theorem 4 does not require the updates to stay in a small domain near the initialization, so it is beyond the lazy training regime. Importantly, the 'dense' bias observed in $\boldsymbol{\beta}$ is not a direct consequence of our model assumption even though we only assume the diagonal elements to be nonzero. In fact, utilizing the diagonal elements can still lead to sparse features (e.g., see [39]). We believe that this bias in DEQs stems essentially from their network architecture. On the other hand, the current implicit bias holds for GF and GD, whereas other optimization algorithms may induce different implicit biases, which we aim to explore in future work.

Based on our results above, we now provide a concrete example to show the advantages brought by the bias of DEQ in out-of-distribution (OOD) tasks. This is motivated by the fact that diversifying spurious features can improve OOD generalization [43]. Specifically, we focus on generalization on the unseen domain (GOTU) setting [36], a rather strong case of OOD generalization where part of the distribution domain is unseen at training but used at testing. As an example, we here utilize the setting in Theorem 3.11 in [36]. Consider the sample space $\mathcal{S} = \{-1, 1\}^d$ and a linear boolean function $f : \mathcal{S} \to \mathbb{R}$ defined as

$$f(\mathbf{x}) = \hat{f}(\emptyset) + \sum_{i=1}^{d} \hat{f}(\{i\})x_i, \tag{13}$$

where $\hat{f}(\{i\}) = \mathbb{E}_{\mathbf{x} \sim_U \{-1,1\}^d}[x_i f(\mathbf{x})]$, $\hat{f}(\emptyset) = \mathbb{E}_{X \sim_U \{-1,1\}^d}[f(\mathbf{x})]$ and $\sim_U \mathcal{U}$ refers to uniform sampling from $\mathcal{U}$. In training, the $k$-th component of every accessible sample is fixed as 1, i.e., the unseen domain is $\mathcal{U} = \{\mathbf{x} \in \{\pm1\}^d : x_k = -1\}$. Denote by $\tilde{f}_{\mathcal{S}\backslash\mathcal{U}}$ the function learned on $\mathcal{S}\backslash\mathcal{U}$. The GOTU error is the defined as the generalization completely on the unseen domain, i.e.,

$$GOTU(f, \tilde{f}, \mathcal{U}) = \mathbb{E}_{X \sim_U \mathcal{U}}[l(\tilde{f}_{\mathcal{S}\backslash\mathcal{U}}(X), f(X))],$$

where $l$ is the quadratic loss function. It is shown in [36] that learning this function with diagonal linear network results in a GOTU error of $4\hat{f}(\{k\})^2 + O(\varepsilon)$ for an infinitesimal $\varepsilon$. On the other hand, the following proposition shows that under mild conditions, learning such function with DEQ achieves smaller GOTU error, where we consider DEQ in Eq. (9) with a bias term, i.e., $f(\mathbf{w}, \mathbf{x}) = \sum_{i=1}^{d} \frac{1}{1-w_i} x_i + b$.

**Proposition 2.** *Let $f(\mathbf{x})$ be defined as in Eq. (13). Assume that*
$$\hat{f}(\{i\}) > 0, \quad \forall 1 \leq i \leq d, \quad \hat{f}(\{k\}) > 1, \quad |\hat{f}(\emptyset)| \leq 2|\hat{f}(\{k\})|.$$
*Consider learning $f$ using gradient flow on population loss[2] on a linear diagonal DEQ with bias initialized by $w_i(0) = b(0) = 0$ for all $i$ with unseen domain $\mathcal{U} = \{\mathbf{x} \in \{\pm1\}^d : x_k = -1\}$. Then the loss converges to 0, and it holds for the generalization error on the unseen that*

$$GOTU \leq 4\left(\hat{f}(\{k\}) - \left(4 + 3\hat{f}(\{k\})\right)^{-\frac{1}{3}}\right)^2 < 4\hat{f}(\{k\})^2.$$

In this setting, the function $\mathbf{x} \mapsto x_k$ has a higher frequency component (i.e., degree) compared to the constant function 1. Consequently, the inductive bias of DEQ enables the model to capture some information about the high-frequency components. We further conduct experiments to study the potential advantages of DEQ in learning high-frequency components in Appendix B.2.

---

[2]It is identical to the setting in Theorem 3.11, [36]. Note that optimizing the population loss in generalization cannot reduce the OOD error.

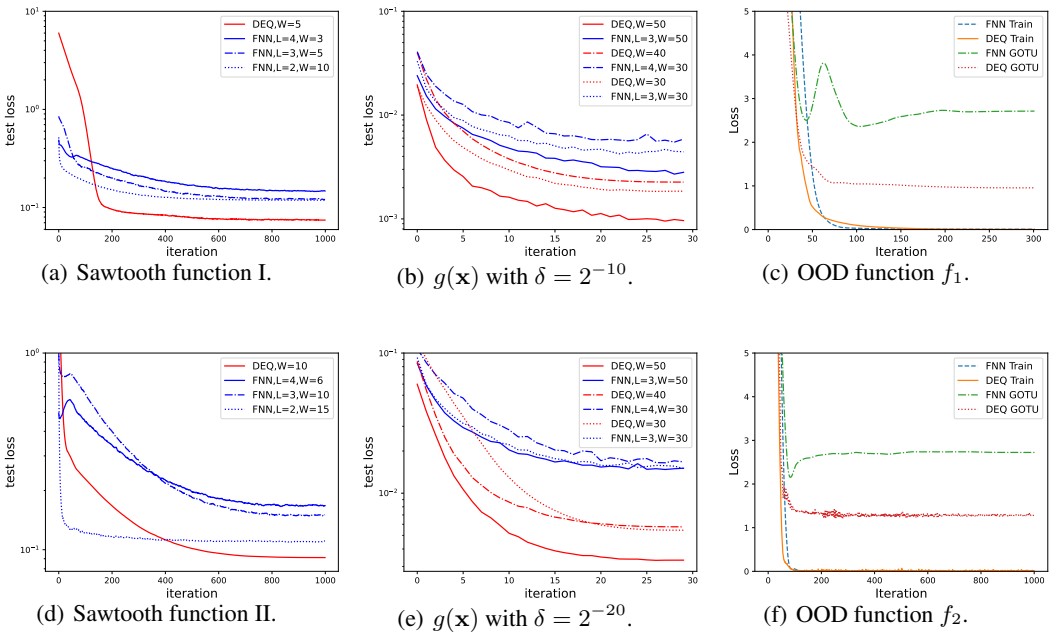

| (a) Sawtooth function I. | (b) $g(\mathbf{x})$ with $\delta = 2^{-10}$. | (c) OOD function $f_1$. |
|---|---|---|
| (d) Sawtooth function II. | (e) $g(\mathbf{x})$ with $\delta = 2^{-20}$. | (f) OOD function $f_2$. |

Figure 1: Test losses of FNN and DEQ networks with various width $W$ and depth $L$. (a) and (d) apply Sawtooth function I and II with $2^5$ and $2^{10}$ linear regions, respectively. (b) and (e) apply function $g(\mathbf{x})$ defined in Eq. (5) with $\delta = 2^{-10}$ and $\delta = 2^{-20}$, respectively. (c) and (f) show the train loss and the GOTU error of FNN and DEQ on the boolean function $f_1$, $f_2$ with unseen domain given by Eq. (14) and Eq. (15).

## 6  Experiments

In this section, we conduct experiments on FNNs and DEQs based on our theoretical results. We first evaluate the expressivity of both networks on the functions proposed in our two separation results. Then we experiment on specific OOD tasks. Several additional experiments on audio representation and mutiscale DEQ are provided in Appendix B.2.

**Piecewise functions.** We first verify the results in Section 4.1. The target function is designed as a saw-tooth function, as defined in Lemma 4 in Appendix A.1, which can be exactly computed by a DEQ. We set the number of linear regions of the saw-tooth function to $2^5$ and $2^{10}$ and experiments on other sawtooth functions can be seen in Appendix B.1. According to Proposition 1, a DEQ with width 5 and 10 can compute the above functions.

Figure 1(a) and Figure 1(d) show that DEQ can achieve nearly zero test loss, demonstrating the saw-tooth function with $2^m$ linear regions can be computed by DEQ. On the other hand, a not-so-deep and not-so-wide FNN fails to achieve test loss as low as DEQ, thus verifying the separation results between FNN and DEQ.

**Solution to quadratic optimization problem.** We then validate the ability of DEQ to approximate the solution function to certain optimization problems. We empirically demonstrate that DEQ can approximate such function better than an FNN with a similar number of parameters. We consider the objective function $g(\mathbf{x})$ defined in Eq. (6), with the input dimension $d$ 10 and 20, and thus $\delta$ in target function being $2^{-10}$ and $2^{-20}$. The input space is $\mathbf{x} \in [0, 1]^d$ with the sampling distribution $p(\mathbf{x}) = \frac{1}{2(1-\delta)}$ for $0 < x_1 < 1 - \delta$ and $p(\mathbf{x}) = \frac{1}{2\delta}$ for $1 - \delta < x_1 < 1$. To verify results under different network parameters, we adjust the layer number and hidden dimension of FNN and the layer width of DEQ while keeping the total number of parameters of both networks similar.

As shown in Figure 1(b) and Figure 1(e), for different network parameters and target functions, DEQ consistently achieves a lower test loss than FNN, demonstrating the superiority of DEQ to approximate and represent functions as solutions to certain optimization problems.

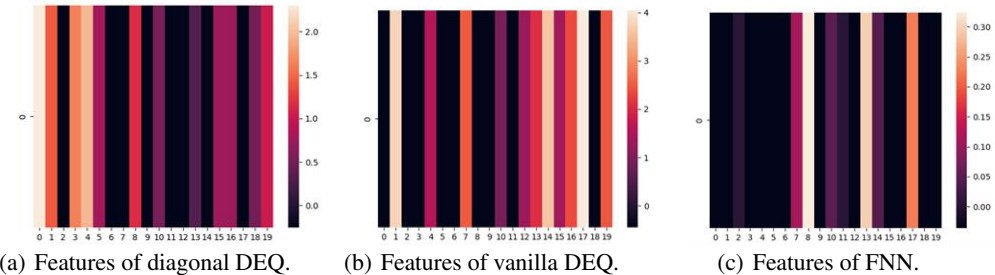

|  (a) Features of diagonal DEQ. | (b) Features of vanilla DEQ. | (c) Features of FNN. |

Figure 2: The heatmaps of diagonal DEQ, vanilla DEQ and FNN. We dispaly the magnitude of feature $z$ of DEQ and the magnitude of feature before the fully-connented layer of FNN. The x-axis represents features 1-20 and darker colors indicate smaller features.

**Out-of-Distribution tasks.** We further perform experiments on the implicit bias of DEQ to verify the advantage of DEQ on OOD tasks. We consider 2 linear boolean functions $f : \mathcal{S} \to \mathbb{R}$ in the form of Eq. (13) and unseen domains $\mathcal{U} \subset \{\pm 1\}^d$. The first function is an example of the mean function and the second function is a part of DTFT. Experiments on other OOD functions can be found in Appendix B.1.

$$f_1(x) = 1.25x_0 + 1.25x_1 + 1.25x_2 + \cdots + 1.25x_{10}, \quad \mathcal{U} = \{\mathbf{x} \in \{\pm 1\}^{10} : x_2 = -1\}, \quad (14)$$

$$f_2(x) = \sum_{n=0}^{9} \sin\left(\frac{\pi n}{10}\right) x_n, \quad \mathcal{U} = \{\mathbf{x} \in \{\pm 1\}^{10} : x_1 = -1\} \quad (15)$$

For each experiment, we generate all binary sequences in $\{\pm 1\}^d \backslash \mathcal{U}$ for training. We employ AdamW optimizer with $\ell_2$ loss and a cosine annealing scheduler. We can observe in Figure1 (c) that the GOTU error of $f_1$ is below the threshold of generalization error based on the Proposition 2. As shown in Figure1 (c) and Figure1 (f), the training loss converges to 0 and the generalization error on the unseen is bounded, which empirically demonstrates the advantage of DEQ on OOD tasks.

In Figure 2, we display the heatmap of the features of diagonal DEQ, vanilla DEQ and FNN trained on the target function with the unseen domain defined in Eq.(14). As the darker colors indicate smaller magnitude of features, we can confirm that the learned features of DEQ are indeed 'denser' than those of FNN.

## 7 Conclusions and Future Directions

In this paper, we provide two separations of DEQ and FNN and analyze the bias of DEQ through the lens of learning dynamics. Our theoretical results provably show the advantage of DEQ over FNN in specific problems and quantify certain learning properties of DEQ. Overall, we conjecture that DEQ may be advantageous in learning certain high-frequency components.

There are many directions that remain to study. First, it is expecting to study the expressivity and bias of more general DEQs. Second, extending our bias analysis under gradient methods to stochastic methods is intriguing. Additionally, it remains promising to incorporate the advantages of DEQ in commonly used networks in real applications. Besides, we also aim to explore DEQ's potential in representing algorithms for reasoning or in-context learning, as evidence suggests the learned networks perform algorithms in some of these tasks.

## Acknowledgements

C. Fang and Z. Lin were supported by National Key R&D Program of China (2022ZD0160300). C. Fang was also supported by the NSF China (No. 62376008). Z. Lin was also supported and the NSF China (No. 62276004).

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

# A Proofs and Discussions

## A.1 Proofs in Subsection 4.1

In following technical lemma, we show that there exists a width-$m$ ReLU-DEQ computing a function with $2^m$ linear regions.

**Lemma 4.** *Let $m \in \mathbb{N}^+$. For all $m \geq 1$, consider the following function on $[0,1]^d$:*

$$\phi^{(m)}(\mathbf{x}) = \begin{cases} 2^m x_1 - 2i, & x_1 \in \left[\frac{2i}{2^m}, \frac{2i+1}{2^m}\right], & 0 \leq i \leq 2^{m-1} - 1, \\ -2^m x_1 + 2i + 2, & x_1 \in \left[\frac{2i+1}{2^m}, \frac{2i+2}{2^m}\right], & 0 \leq i \leq 2^{m-1} - 1. \end{cases}$$

*Then there exists a DEQ with width $m$ that exactly computes $-\phi^{(m)}(\mathbf{x}) + 2^m x_1$ on $[0,1]^d$. Moreover, the DEQ has $2^m$ linear regions on $[0,1]^d$.*

*Proof.* Since $2^m x_1$ is a linear function with respect to $z_1$, by definition, $-\phi^{(m)}(\mathbf{x}) + 2^m x_1$ has 2 linear regions on $\left[\frac{2i}{2^m}, \frac{2i+2}{2^m}\right] \times [0,1]^{d-1}$ for all $0 \leq i \leq 2^{m-1} - 1$. Thus it has $2^m$ linear regions on $[0,1]^d$. It suffices to show that existence of a DEQ computing $-\phi^{(m)}(\mathbf{x}) + 2^m x_1$.

Consider a width-$m$ DEQ with weight matrices as follows:

$$\mathbf{A}^T = \begin{pmatrix} -2^m \\ -2^{m-1} \\ \vdots \\ -2 \end{pmatrix}, \mathbf{W} = \begin{pmatrix} 0 & & & \\ -4 & 0 & & \\ -8 & -4 & 0 & \\ \vdots & \vdots & \vdots & \ddots \\ -2^m & -2^{m-1} & -2^{m-2} & \cdots & 0 \end{pmatrix}, \mathbf{U}_1 = \begin{pmatrix} 2 \\ 4 \\ \vdots \\ 2^m \end{pmatrix}, \mathbf{b} = \begin{pmatrix} -1 \\ -1 \\ \vdots \\ -1 \end{pmatrix},$$

where $\mathbf{U}_1$ denotes the first column of $\mathbf{U}$ and $\mathbf{U} = (\mathbf{U}_1 \quad \mathbf{0})$. When a DEQ is equipped with the above weight matrices, direct calculations show that the fixed point $\mathbf{z}$ satisfy

$$z_1(\mathbf{x}) = \sigma(2x_1 - 1), \quad z_t(\mathbf{x}) = \sigma\left(-\sum_{i=1}^{t-1} 2^{t-i+1} z_i(\mathbf{x}) + 2^t x_1 - 1\right), \quad \forall 2 \leq t \leq m. \tag{16}$$

Note that $\{\phi^{(m)}(\mathbf{x})\}$ admits a recursive expression:

$$\phi^{(m+1)}(\mathbf{x}) = 2\phi^{(m)}(\mathbf{x}) - 2\sigma(2\phi^{(m)}(\mathbf{x}) - 1), \quad \forall m \geq 0 \tag{17}$$

for $\phi^{(0)}(\mathbf{x}) := x_1$. We now show by induction that $z_t(\mathbf{x}) = \sigma(2\phi^{(t-1)}(x) - 1)$ for all $1 \leq t \leq m$. When $t = 1$, it is true immediately from Eq. (16) and (17). Assume it is true for some $t < m$, then by Eq. (16) we have

$$\begin{aligned} z_{t+1}(\mathbf{x}) &= \sigma\left(\sum_{i=1}^{t} -2^{t-i+2} z_i(x) + 2^{t+1} x_1 - 1\right) \\ &= \sigma\left(\sum_{i=1}^{t} -2^{t-i+2} \sigma(2\phi^{(i-1)}(x) - 1) + 2^{t+1} x_1 - 1\right) \\ &= \sigma\left(\sum_{i=1}^{t} -2^{t-i+2}\left(\phi^{(i-1)}(x) - \frac{\phi^{(i)}(x)}{2}\right) + 2^{t+1} x_1 - 1\right) \\ &= \sigma(-2^{t+1}\phi^{(0)}(x) + 2\phi^{(t)}(x) + 2^{t+1} x_1 - 1) = \sigma(2\phi^{(t)}(x) - 1), \end{aligned}$$

where we use the induction in the second line, Eq. (17) in the third line, and $\phi^{(0)}(\mathbf{x}) = x_1$ in the last line. Thus the induction holds.

Using the induction and Eq. (17) for the DEQ, we have

$$
\mathbf{A}\,\mathbf{z}(\mathbf{x}) = \sum_{i=1}^{m} -2^{m+1-i} z_i(\mathbf{x})
$$

$$
= \sum_{i=1}^{m} -2^{m+1-i} \sigma(2\phi^{(i-1)}(\mathbf{x}) - 1)
$$

$$
= \sum_{i=1}^{m} -2^{m+1-i} \left( \phi^{(i-1)}(\mathbf{x}) - \frac{\phi^{(i)}(\mathbf{x})}{2} \right)
$$

$$
= -2^m \phi^{(0)}(\mathbf{x}) + \phi^{(m)}(\mathbf{x}) = \phi^{(m)}(\mathbf{x}) - 2^m x_1,
$$

and the lemma follows. $\qquad\square$

To prove the theorem, we also need the following lemma which is proved in [20].

**Lemma 5** (Lemma 2.1 in [20]). *Let $k \in \mathbb{N}^+$, $L \geq 2$ and $\rho(x) : \mathbb{R} \to \mathbb{R}$ be a piecewise affine linear function with $p$ pieces. Then every $f : \mathbb{R} \to \mathbb{R}$ implemented by an FNN with depth $L$, width $k$ and activation function $\rho$ has at most $(pk)^{L-1}$ linear regions.*

Note that the in Lemma 4, the function computed by DEQ is a variant of the saw-tooth function that has many linear regions. On the other hand, Lemma 5 provides an upper bound on the number of linear regions generated by FNN. Combining these two lemmas and using a technique similar to that in Theorem 1.1 in [24], we are able to prove Theorem 1.

*Proof of Theorem 1.* Let $N_d(\mathbf{x})$ be the DEQ in Lemma 4 that computes $2^m x_1 - \phi^{(m)}(\mathbf{x})$ and denote the width of the FNN that computes $N_f(x)$ by $k$. For any $\mathbf{y} \in [0,1]^{d-1}$, define $p_\mathbf{y}(x) : [0,1] \to [0,1]^d$ as $p_\mathbf{y} = (x_1, \mathbf{y})$. Then for $N_f \circ p_\mathbf{y}(x) : [0,1] \to \mathbb{R}$, by Lemma 5, the number of linear regions is upper bounded by

$$
(pk)^{L-1} \leq 2^{\left(m^{1-\alpha}-1\right)(L-1)} \leq 2^{m-2},
$$

where $p = 2$ denotes the number of linear pieces of ReLU activation function. Therefore, $N_f \circ p_\mathbf{y}(x) - 2^m x$ has at most $2^{m-2}$ linear regions on $[0,1]$.

Note that $\phi^{(m)}(\mathbf{x})$ only depends on the first entry of $\mathbf{x}$, for simplicity, we define $\varphi^{(m)}(x) : \mathbb{R} \to \mathbb{R}$ as $\varphi^{(m)}(x) = \phi^{(m)} \circ p_\mathbf{y}(x)$. Now we claim that there exists at least $3 \cdot 2^{m-3} - 2$ small intervals $\{T_l\}_l$ on $[0,1]$ with $\mathrm{diam}(T_l) = 2^{-m}$, such that for any $\mathbf{y}$, it holds that

$$
\mathrm{sgn}\left( \varphi^{(m)}(x) - \frac{1}{2} \right) \neq \mathrm{sgn}\left( N_f \circ p_\mathbf{y}(x) - 2^m x - \frac{1}{2} \right), \quad \forall x \in T_l, \quad \forall l.
$$

For simplicity, denote $\tilde{\varphi}(x) = \varphi^{(m)}(x) - \frac{1}{2}$ and $\tilde{N}_f(x) = N_f \circ p_\mathbf{y}(x) - 2^m x - \frac{1}{2}$. Denote $\mathcal{P}_\phi$ and $\mathcal{P}_N$ the partitions of $[0,1]$ into intervals so that $\mathrm{sgn}(\tilde{\varphi}(x))$ and $\mathrm{sgn}(\tilde{N}_f(x))$ remains constant within each interval, respectively. Let $\mathcal{I}_\phi$ be the set of all intervals partitioned by $\mathcal{P}_\phi$ and $\mathcal{I}_N$ be the set of all intervals partitioned by $\mathcal{P}_N$. By definition, $|\mathcal{I}_\phi| = 2^m + 1$. Since $\tilde{N}_f(x)$ has at most $2^{m-2}$ linear regions, the number of the boundary points of the intervals in $\mathcal{I}_N$ is upper bounded $2^{m-2} + 1$. So there are at least $2^m + 1 - (2^{m-2} + 1) = 3 \cdot 2^{m-2}$ intervals in $\mathcal{I}_\phi$ that do not intersect with any boundary points of intervals, i.e., lie completely in an interval in $\mathcal{I}_N$. Denote this set of intervals by $\mathcal{I}_\phi'$. On the other hand, since $\mathrm{sgn}(\tilde{N}_f(x))$ remains constant on every interval in $\mathcal{I}_N$, for every $J \in \mathcal{I}_N$ that contains $i_J$ intervals in $\mathcal{I}_\phi'$, there will be $\frac{i_J+1}{2}$ intervals when $i_J$ is odd and $\frac{i_J}{2}$ intervals when $i_J$ is even, on which $\mathrm{sgn}(\tilde{\varphi}(x)) = \mathrm{sgn}(\tilde{N}_f(x))$. Note that $\sum_{J \in \mathcal{I}_N} i_J \geq 3 \cdot 2^{m-2}$. Therefore, among the sets in $\mathcal{I}_\phi'$, the number of sets on which $\mathrm{sgn}(\tilde{\varphi}(x)) \neq \mathrm{sgn}(\tilde{N}_f(x))$ is at least

$$
3 \cdot 2^{m-2} - \sum_{J \in \mathcal{I}_N} \frac{i_J + 1}{2} \geq 2^{m-3},
$$

where we use the fact that $|\mathcal{I}_N| \leq 2^{m-2} + 1$. Note that except for two intervals, every $T \in \mathcal{I}_\phi'$ can be represented as $\left[ \frac{4i+1}{2^{m+1}}, \frac{4i+3}{2^{m+1}} \right]$ or $\left[ \frac{4i-1}{2^{m+1}}, \frac{4i+1}{2^{m+1}} \right]$ for some $i$, thus $\mathrm{diam}(T_l) = 2^{-m}$, which proves the

claim. Moreover, on each $T_l$, direct calculations show $\int_{T_l} \left| \phi^{(m)}(x) - \frac{1}{2} \right| \mathrm{d}x \geq 2^{-m-2}$. Therefore, by using the claim, we have

$$
\int_{[0,1]^d} |N_d(\mathbf{x}) - N_f(\mathbf{x})| \mathrm{d}\mathbf{x}
$$

$$
= \int_{[0,1]^{d-1}} \int_{[0,1]} \left| 2^m x_1 - \phi^{(m)}(x_1) - N_f \circ p_\mathbf{y}(x_1) \right| \mathrm{d}x_1 \mathrm{d}\mathbf{y}
$$

$$
\geq \int_{[0,1]^{d-1}} \int_{\bigcup_l T_l} \left| 2^m x_1 - \phi^{(m)}(x_1) - N_f \circ p_\mathbf{y}(x_1) \right| \mathrm{d}x_1 \mathrm{d}\mathbf{y}
$$

$$
\geq \int_{[0,1]^{d-1}} \int_{\bigcup_l T_l} \left| \frac{1}{2} - \phi^{(m)}(x_1) \right| \mathrm{d}x_1 \mathrm{d}\mathbf{y}
$$

$$
\geq \int_{[0,1]^{d-1}} |T_l| \cdot 2^{-m-2} \mathrm{d}\mathbf{y}
$$

$$
\geq (3 \cdot 2^{m-3} - 2) \cdot 2^{-m-2} \geq \frac{1}{16}.
$$

$\square$

Next we turn to proof Proposition 1. We use $\mathrm{Diag}(\cdot)$ to extract the diagonal elements of a matrix into a vector. The proof of Proposition 1 relies on following explicit expression of ReLU DEQ.

**Lemma 6.** *Let* $\mathbf{W}, \mathbf{U}, \mathbf{b}$ *be the weights of a DEQ with* $\| \mathbf{W} \|_2 < 1$. *Then for any* $\mathbf{x} \in \mathbb{R}^d$, *there exists a diagonal matrix* $\mathbf{D} \in \mathbb{R}^{d \times d}$ *whose diagonal entries are either* 1 *or* 0, *such that*

$$
\mathrm{sgn}(\mathrm{diag}((\mathbf{I} - \mathbf{W}\mathbf{D})^{-1})(\mathbf{U}\mathbf{x} + \mathbf{b})) = \mathrm{Diag}(\mathbf{D}). \tag{18}
$$

*Moreover, fix* $\mathbf{D}$, *for all* $\mathbf{x}$ *that Eq.* (18) *holds, we have*

$$
\mathbf{z}(\mathbf{x}) = (\mathbf{I} - \mathbf{D}\mathbf{W})^{-1}\mathbf{D}(\mathbf{U}\mathbf{x} + \mathbf{b}). \tag{19}
$$

*Proof.* Recall that the fixed point $\mathbf{z}(\mathbf{x})$ satisfies

$$
\mathbf{z} = \sigma(\mathbf{W}\mathbf{z} + \mathbf{U}\mathbf{x} + \mathbf{b}). \tag{20}
$$

For each $z_i$, if the $i$-th entry of $(\mathbf{W}\mathbf{z} + \mathbf{U}\mathbf{x} + \mathbf{b})$ is smaller than 0, then $z_i = 0$. Without loss of generality, we assume that the first $t$ $(t \leq m)$ entries of $(\mathbf{W}\mathbf{z} + \mathbf{U}\mathbf{x} + \mathbf{b})$ are greater than 0, and the rest $m - t$ entries are smaller than 0. Denote by $\mathbf{v} = \mathbf{U}\mathbf{x} + \mathbf{b}$ and the corresponding block matrices $\mathbf{z}, \mathbf{W}, \mathbf{v}$ by

$$
\mathbf{z} = \begin{pmatrix} \tilde{\mathbf{z}} \\ \mathbf{0} \end{pmatrix}, \mathbf{W} = \begin{pmatrix} \mathbf{W}_{11} & \mathbf{W}_{12} \\ \mathbf{W}_{21} & \mathbf{W}_{22} \end{pmatrix}, \mathbf{v} = \begin{pmatrix} \mathbf{v}_1 \\ \mathbf{v}_2 \end{pmatrix}, \tag{21}
$$

where $\tilde{\mathbf{z}} \in \mathbb{R}^t, \mathbf{W}_{11} \in \mathbb{R}^{t \times t}$, and $\mathbf{v}_1 \in \mathbb{R}^t$. Then, Eq.(20) is equivalent to

$$
\tilde{\mathbf{z}} = \mathbf{W}_{11}\tilde{\mathbf{z}} + \mathbf{v}_1, \quad \mathbf{W}_{21}\tilde{\mathbf{z}} + \mathbf{v}_2 \leq 0, \quad \tilde{\mathbf{z}} > 0. \tag{22}
$$

Now we define $\mathbf{D} = \begin{pmatrix} \mathbf{I}_t & \mathbf{0} \\ \mathbf{0} & \mathbf{0} \end{pmatrix}$ and show that it is the desired matrix. Note that $\| \mathbf{W} \|_2 < 1$ and $\| \mathbf{D} \|_2 = 1$, we have

$$
\| \mathbf{W}_{11} \|_2 = \| \mathbf{W}\mathbf{D} \|_2 \leq \| \mathbf{W} \|_2 \| \mathbf{D} \|_2 < 1,
$$

showing that $\mathbf{I}_t - \mathbf{W}_{11}$ is invertible. Thus Eq.(22) gives

$$
\tilde{\mathbf{z}} = (\mathbf{I}_t - \mathbf{W}_{11})^{-1}\mathbf{v}_1 > 0, \quad \mathbf{W}_{21}(\mathbf{I}_t - \mathbf{W}_{11})^{-1}\mathbf{v}_1 + \mathbf{v}_2 \leq 0. \tag{23}
$$

Additionally, by simple calculation, we have

$$
(\mathbf{I} - \mathbf{D}\mathbf{W})^{-1} = \begin{pmatrix} (\mathbf{I}_t - \mathbf{W}_{11})^{-1} & (\mathbf{I}_t - \mathbf{W}_{11})^{-1}\mathbf{W}_{12} \\ \mathbf{0} & \mathbf{I} \end{pmatrix},
$$

$$
(\mathbf{I} - \mathbf{W}\mathbf{D})^{-1} = \begin{pmatrix} (\mathbf{I}_t - \mathbf{W}_{11})^{-1} & \mathbf{0} \\ \mathbf{W}_{21}(\mathbf{I}_t - \mathbf{W}_{11})^{-1} & \mathbf{I} \end{pmatrix}. \tag{24}
$$

Combining Eq. (21), (23) and (24), we have

$$(\mathbf{I} - \mathbf{W}\,\mathbf{D})^{-1}(\mathbf{U}\,\mathbf{x} + \mathbf{b}) = \begin{pmatrix} \mathbf{W}_{11}\tilde{\mathbf{z}} + \mathbf{v}_1 \\ \mathbf{W}_{21}(\mathbf{I}_t - \mathbf{W}_{11})^{-1}\mathbf{v}_1 + \mathbf{v}_2 \end{pmatrix},$$

$$\mathbf{z} = \begin{pmatrix} (\mathbf{I}_t - \mathbf{W}_{11})^{-1}\mathbf{v}_1 \\ \mathbf{0} \end{pmatrix} = (\mathbf{I} - \mathbf{D}\mathbf{W})^{-1}\mathbf{D}(\mathbf{U}\,\mathbf{x} + \mathbf{b}).$$

Finally, since the output of the implicit layer is unique, in the sense of permuting the entries of $\mathbf{D}$, there always exists a matrix $\mathbf{D}$ such that the lemma follows. $\qquad\square$

Note that there are at most $2^m$ diagonal matrix whose diagonal entries are either 1 or 0, the upper bound of the number of linear regions is $2^m$. Thus Proposition 1 follows straightforwardly from Lemma 6 and 4.

### A.2 Proofs in Subsection 4.2

#### A.2.1 Inapproximability of FNNs

The goal of this section is to prove the following proposition, which is an extended version of the inapproximability result in Theorem 2.

**Assumption 2.** *The activation function $\tilde{\sigma}$ is of $\mathcal{C}^0(\mathbb{R})$ and continuous differentiable except for at most finitely many points. And there exists an absolute constant $C_{\tilde{\sigma}} > 0$, such that $|\tilde{\sigma}'(x)| \leq C_{\tilde{\sigma}}$ for all $x$ on which $\tilde{\sigma}$ is differentiable.*

**Proposition 3** (Inapproximability of FNN). *Let $N_{fnn}(x)$ be computed by an FNN with depth $L$, width $k$, and an activation function $\tilde{\sigma}$ satisfying Assumption 2 on $\mathbf{x} \in [0, 1]^d$. Let $g(\mathbf{x})$ be defined as in Eq.(6), and $\frac{1}{4} \geq \varepsilon > 0$. If $\|N_{fnn}(\mathbf{x}) - g(\mathbf{x})\|_{L^\infty([0,1]^d)} \leq \varepsilon$, then there exists a weight parameter $W_{ij}$ of the FNN for $1 \leq i \leq L$ and $1 \leq j \leq k$, such that*

$$|W_{ij}| \geq \frac{1}{C_{\tilde{\sigma}}k} \cdot 2^{\frac{d-4}{L}}.$$

*Proof.* By assumption, $N_{\mathrm{fnn}}(\mathbf{x})$ is of $\mathcal{C}^0(\mathbb{R}^d)$ and continuous differentiable except for at most finitely many points, then by the intermediate value theorem, we have

$$\max_{x \in [0,1]^d} \left| \frac{\partial N_{\mathrm{fnn}}(\mathbf{x})}{\partial x_1} \right| \geq \left| \frac{g_1(1) - g_1(1 - \delta)}{\delta} \right| \geq \frac{\frac{1}{2} - \frac{\delta(1-\delta)}{4\delta} - 2 \cdot \frac{1}{16}}{\delta} \geq \frac{1}{8\delta} - 1 \geq 2^{d-4}, \quad (25)$$

where $\frac{\partial N_{\mathrm{fnn}}(\mathbf{x})}{\partial x_1}$ refers to the subgradient on the non-differentiable points. Additionally, by definition,

$$N_{\mathrm{fnn}}(\mathbf{x}) = \mathbf{W}_L \tilde{\sigma}\left(\mathbf{W}_{L-1}\tilde{\sigma}(\cdots \tilde{\sigma}(\mathbf{W}_1\,\mathbf{x} + \mathbf{b}_1)\cdots) + \mathbf{b}_{L-1}\right)$$

Then direct calculation gives

$$\nabla N_{\mathrm{fnn}}(\mathbf{x}) = \mathbf{W}_1^T \mathbf{D}_1 \cdots \mathbf{W}_{L-1}^T \mathbf{D}_{L-1}\mathbf{W}_L^T, \quad (26)$$

where $\mathbf{D}_l = \mathrm{diag}(\tilde{\sigma}'(\mathbf{W}_l\tilde{\sigma}(\cdots \tilde{\sigma}(\mathbf{W}_1\mathbf{x} + \mathbf{b}_1)\cdots) + \mathbf{b}_l))$ for $1 \leq l \leq L - 1$. By Assumption 2, it holds that

$$\|\mathbf{D}_l\|_\infty \leq C_{\tilde{\sigma}}, \quad \forall 1 \leq l \leq L - 1.$$

Then combining Eq. (25) and (26), we have

$$2^{d-4} \leq \|\nabla N_{\mathrm{fnn}}(\mathbf{x})\|_\infty \leq \prod_{i=1}^{L} \|\mathbf{D}_i \mathbf{W}_i\|_\infty \leq C_{\tilde{\sigma}}^L \cdot \prod_{i=1}^{L} \|\mathbf{W}_i\|_\infty.$$

Therefore, there exists at least one $\mathbf{W}_i$ for $1 \leq i \leq L$, such that

$$\|\mathbf{W}_i\|_\infty \geq C_{\tilde{\sigma}}^{-1} 2^{\frac{d-4}{L}}.$$

Finally, by the definition of $\|\cdot\|_\infty$, there exists an entry $W_{ij}$ with $1 \leq j \leq k$, such that

$$|W_{ij}| \geq \frac{1}{C_{\tilde{\sigma}}k} \cdot 2^{\frac{d-4}{L}}.$$

$\qquad\square$

**Remark 5.** *Assumption 2 is mild and one can verify that most commonly used activation functions such as ReLU, GeLU, sigmoid and tanh satisfy the assumption.*

To prove the inapproximability of FNNs in Theorem 2, we take $C_{\tilde{\sigma}} = 1$ in Proposition 3 as $|\sigma'(x)| \leq 1$ and derive

$$|W_{ij}| \geq k^{-1} \cdot 2^{\frac{d-4}{L}} \leq 2^{-\frac{d}{2C} + \frac{d-4}{C}} \geq 2^{-\frac{d}{2C}},$$

which finishes the proof.

### A.2.2 Approximability of DEQs

This section centers around the approximability result of DEQs. We restate the approximability result of Theorem 2 as the following proposition.

**Proposition 4** (Approximability of DEQ). *Let $g(\mathbf{x})$ be defined as in Eq.(6) on $[0,1]^d$. $\forall \frac{1}{4} \geq \varepsilon > 0$, there exists a DEQ $N_{deq}$ with width bounded by $5\varepsilon^{-1}$ and weights bounded by $2\varepsilon^{-1}$, such that*

$$\|N_{deq}(\mathbf{x}) - g(\mathbf{x})\|_{L^\infty([0,1]^d)} \leq \varepsilon.$$

The proof of the proposition requires some intermediate steps regrading the constructing approximation by DEQ and bounding the fixed-points' error. For simplicity, in the rest of the section, for any function $f$ which is continuous differentiable except for at most finitely many points, we denote $f'$ the derivative of $f$ on the differentiable points, and the subgradient of $f$ on the non-differentiable points.

The next lemma considers approximating the square function using a 2-layer FNN.

**Lemma 7.** *For any $N \in \mathbb{N}^+$, there exists a function $\phi$ implemented by a 2-layer ReLU FNN with width $2N$ such that*

$$|\phi(x) - x^2| \leq 4N^{-2}, \quad |\phi'(x)| \leq 2 - \frac{1}{N}, \quad \forall x \in [-1, 1].$$

*Proof.* Denote $\frac{1}{N}$ by $t$ for simplicity. Let $\{x_i\}_{i=1}^{2N+1}$ be $2N+1$ points on $\mathbb{R}$ be defined as follows:

$$x_1 = -1, \ x_2 = -1 + t, \ \cdots, \ x_{2N} = 1 - t, \ x_{2N+1} = 1.$$

We consider the following function $\phi(x)$ that interpolates $x^2$ on all $\{x_i\}_{i=1}^{2N+1}$:

$$\phi(x) = \sigma(tx) + \sigma(-tx) + \sum_{i=1}^{N-1} \sigma(2tx - 2it^2) + \sum_{i=1}^{N-1} \sigma(-2tx + 2it^2). \tag{27}$$

It can be seen that $\phi(x)$ can be implemented by a 2-layer ReLU FNN with width $2N$ and weight bounded by $2t$. By the interpolation property of $\phi(x)$, on every $[x_j, x_{j+1}]$, it holds

$$\max_{x \in [x_j, x_{j+1}]} |\phi(x) - x^2| = \phi\left(\frac{x_j + x_{j+1}}{2}\right) - \left(\frac{x_j + x_{j+1}}{2}\right)^2 = \frac{t^2}{4}.$$

Thus we have $|\phi(x) - x^2| \leq 4N^{-2}$ for all $x \in [-1, 1]$. Moreover, since $\phi(x)$ is convex, we have

$$|\phi'(x)| \leq \frac{1 - (1-t)^2}{t} = 2 - t.$$

$\square$

We now move to prove the equivalence between the revised DEQ and vanilla DEQ.

*Proof of Lemma 1.* For any $\hat{\mathbf{z}}^0 \in \mathbb{R}^m$, we define a sequence $\{\hat{\mathbf{z}}^k\}$ as

$$\hat{\mathbf{z}}^{k+1} = \sigma(\mathbf{W}\,\mathbf{V}\hat{\mathbf{z}}^k + \mathbf{U}\mathbf{x} + \mathbf{b}).$$

Since $\|\mathbf{W}\,\mathbf{V}\|_2 \leq 1$, $\{\hat{\mathbf{z}}^k\}$ converges and the limit $\hat{\mathbf{z}}^*$ is the fixed point of $\hat{\mathbf{z}} = \sigma(\mathbf{W}\mathbf{V}\hat{\mathbf{z}} + \mathbf{U}\mathbf{x} + \mathbf{b})$. Now we set $\mathbf{z}^0 = \mathbf{V}\mathbf{y}^0$ and define another sequence $\{\mathbf{z}^k\}$ as

$$\mathbf{z}^{k+1} = \mathbf{V}\sigma(\mathbf{W}\mathbf{z}^k + \mathbf{U}\mathbf{x} + \mathbf{b}), \quad \forall k \geq 0.$$

It follows immediately by induction that $\mathbf{z}^k = \mathbf{V}\hat{\mathbf{z}}^k$ for all $k \geq 0$. Note that $\{\mathbf{z}^k\}$ converges and the limit $\mathbf{z}^*$ is exactly the fixed point of the revised DEQ in Eq. (8). Therefore, it holds that

$$\mathbf{z}^* = \lim_{k \to \infty} \mathbf{z}^k = \lim_{k \to \infty} \mathbf{V}\hat{\mathbf{z}}^k = \mathbf{V}\hat{\mathbf{z}}^*.$$

The desired DEQ is constructed as

$$\hat{\mathbf{z}} = \sigma(\mathbf{W}\,\mathbf{V}\hat{\mathbf{z}} + \mathbf{U}\,\mathbf{x} + \mathbf{b}),$$
$$\hat{\mathbf{y}} = (\mathbf{B}\mathbf{V})\hat{\mathbf{z}}$$

$\square$

In the following we turn to bound the error between the equilibria of two fixed-point equations. We start with the proof of Lemma 2.

*Proof of Lemma 2.* The existence of $z_u$ and $z_v$ follows from the fixed point theorem since $u(\cdot, \mathbf{x})$ and $v(\cdot, \mathbf{x})$ are contraction mappings. For simplicity, we denote $u_{\mathbf{x}}(z) = u(z, \mathbf{x})$ and $v_{\mathbf{x}}(z) = v(z, \mathbf{x})$. Note that the range of $u_{\mathbf{x}}$ and $v_{\mathbf{x}}$ are in $\Omega$. Then $\forall n \in \mathbb{N}^+$, we have

$$
\begin{aligned}
\|u_{\mathbf{x}}^{\circ n} - v_{\mathbf{x}}^{\circ n}\|_\infty &\leq \left\|u_{\mathbf{x}}^{\circ n} - u_{\mathbf{x}}\left(v_{\mathbf{x}}^{\circ(n-1)}\right)\right\|_\infty + \left\|u_{\mathbf{x}}\left(v_{\mathbf{x}}^{\circ(n-1)}\right) - v_{\mathbf{x}}^{\circ n}\right\|_\infty \\
&\leq L_u \left\|u_{\mathbf{x}}^{\circ(n-1)} - v_{\mathbf{x}}^{\circ(n-1)}\right\|_\infty + \|u_{\mathbf{x}} - v_{\mathbf{x}}\|_\infty \\
&\leq L_u \left(\left\|u_{\mathbf{x}}^{\circ(n-2)} - v_{\mathbf{x}}^{\circ(n-2)}\right\|_\infty + \|u_{\mathbf{x}} - v_{\mathbf{x}}\|_\infty\right) + \|u_{\mathbf{x}} - v_{\mathbf{x}}\|_\infty \\
&\leq \cdots \\
&\leq (1 + L_u + \cdots + L_u^{n-1})\|u_{\mathbf{x}} - v_{\mathbf{x}}\|_\infty \\
&= \frac{1 - L_u^n}{1 - L_u}\|u_{\mathbf{x}} - v_{\mathbf{x}}\|_\infty.
\end{aligned}
$$

By definition, $\forall (z, \mathbf{x}) \in \Omega \times [0,1]^d$, $z_u(\mathbf{x}) = \lim_{n \to \infty} u_{\mathbf{x}}^{\circ n}(z)$, and $z_v(\mathbf{x}) = \lim_{n \to \infty} v_{\mathbf{x}}^{\circ n}(z)$. Hence, we have

$$
\begin{aligned}
|z_u(\mathbf{x}) - z_v(\mathbf{x})| &\leq \lim_{n \to \infty} |u_{\mathbf{x}}^{\circ n}(z) - v_{\mathbf{x}}^{\circ n}(z)| \\
&\leq \lim_{n \to \infty} \frac{1 - L_u^n}{1 - L_u} \cdot \max_{z \in \Omega} |u(z, \mathbf{x}) - v(z, \mathbf{x})| \\
&\leq \frac{1}{1 - L_u} \cdot \max_{z \in \Omega} |u(z, \mathbf{x}) - v(z, \mathbf{x})|.
\end{aligned}
$$

Finally, by the symmetry of $u$ and $v$, we also have $|z_u(\mathbf{x}) - z_v(\mathbf{x})| \leq \frac{1}{1 - L_v} \cdot \max_{z \in \Omega} |u(z, \mathbf{x}) - v(z, \mathbf{x})|$. The proof is finished. $\square$

We also need Lemma 3 to bound the error.

*Proof of Lemma 3.* We use the intermediate value theorem to proof the lemma. Define $q(z, \mathbf{x}) = z - v(z, \mathbf{x})$. The fixed point $z_v(\mathbf{x})$ is the unique zero of $q(z, \mathbf{x}) = 0$. Since $v(z, \mathbf{x})$ is $L_v$ Lipschitz with respect to $z$ and $L_v < 1$, $q(z, \mathbf{x})$ is monotonically increasing with respect to $z$ for all $\mathbf{x}$.

Fix $z_u$, the proof proceeds by discussing the following 2 cases:

- If $q(z_u, \mathbf{x}) \leq 0$, i.e., $u(z_u, \mathbf{x}) = z_u \leq v(z_u, \mathbf{x})$, we consider $T = [z_u, z_u + \xi] \subset \Omega$. By assumption, there exists $z^* \in T$, such that $u(z^*, \mathbf{x}) = v(z^*, \mathbf{x})$. Note that $q(\cdot, \mathbf{x})$ is monotonically increasing, thus we have $q(z^*, \mathbf{x}) \geq 0$. By the continuity of $q(z, \mathbf{x})$ w.r.t. $z$ and the intermediate value theorem, $q(z, \mathbf{x})$ must have a zero in $[z_u, z_0] \subset T$, which is $z_v(\mathbf{x})$ by definition. Hence, it holds that $|z_u - z_v| \leq \xi$.

- If $q(z_u, \mathbf{x}) \geq 0$, i.e., $u(z_u, \mathbf{x}) = z_u \geq v(z_u, \mathbf{x})$, we consider $T = [z_u - \xi, z_u] \subset \Omega$. It follows from similar deductions that $|z_u - z_v| \leq \xi$ in this case.

We finish the proof. □

With the results above, we begin our formal proof of Proposition 4. The proof is sketched as follows: First, we consider a fixed point equation $z = \tilde{g}(z, \mathbf{x})$ that induces the target function $g(\mathbf{x})$. We show that there exists a function $\tilde{h}(z, \mathbf{x}) : \mathbb{R}^{d+1} \to \mathbb{R}$ computed by a 2-layer FNN with width $O(\varepsilon^{-1})$ that can approximate $\tilde{g}(z, \mathbf{x})$ in sup-norm to an accuracy of $O(\varepsilon^2)$. Moreover, $z = \tilde{h}(z, \mathbf{x})$ is a well-posed fixed point equation and induces a revised DEQ. Second, we bound the error between $g(\mathbf{x})$ and $h(\mathbf{x})$, where $h(\mathbf{x})$ is the fixed point of $z = \tilde{h}(z, \mathbf{x})$. The proof is further divided into two parts: When $1 - x_1 > \frac{\xi}{2}$, by using Lemma 2, we can bound the error $\|h - g\|$ by $\varepsilon \cdot \|\tilde{h} - \tilde{g}\|$. When $1 - x_1 < \frac{\varepsilon}{2}$, we show that the conditions of Proposition 4 holds for $\xi = \varepsilon$, thus $\|h - g\|$ is upper bounded by $\varepsilon$.

*Proof of Proposition 4.* Let $g(\mathbf{x})$ be defined as in Eq.(6). Recall that $g(\mathbf{x})$ is the fixed point of the fixed point equation

$$z = \tilde{g}(z, \mathbf{x}) := z x_1 + \delta \left( \frac{x_1}{2} - z \right).$$

**Approximate $\tilde{g}$ using 2-layer FNN**. By Lemma 7,$\forall N \in \mathbb{N}^+$, there exist $\mathbf{a} \in \mathbb{R}^{2N}, \tilde{\mathbf{b}} \in \mathbb{R}^{2N}, \tilde{\mathbf{W}} \in \mathbb{R}^{2N}$ and a function $\phi(x) = \mathbf{a}^T \sigma(\tilde{\mathbf{W}} x + \tilde{\mathbf{b}})$ , such that for all $x \in [-1, 1]$, it holds

$$|\phi_N(x) - x^2| \leq 4N^{-2}, \quad |\phi_N'(x)| \leq 2 - \frac{1}{N}. \tag{28}$$

Now, we define

$$\tilde{h}(z, \mathbf{x}) = \frac{1}{2} \left[ \phi_N \left( z + \frac{x_1}{2} \right) - \phi_N \left( z - \frac{x_1}{2} \right) \right] + \delta \left( \frac{x_1}{2} - z \right).$$

1. $\tilde{h}(z, \mathbf{x})$ can be implemented by a 2-layer ReLU FNN with width $4N + 2$ for $(z, \mathbf{x}) \in \left[ -\delta, \frac{1}{2} \right] \times [0, 1]^d$. To see this, when $(z, \mathbf{x}) \in \left[ -\delta, \frac{1}{2} \right] \times [0, 1]^d$, it holds $z + \frac{x_1}{2} \in [0, 1]$, $z - \frac{x_1}{2} \in [-1, 0]$. Then

$$
\begin{pmatrix} \frac{\mathbf{a}^T}{2} & \frac{\mathbf{a}^T}{2} & -\delta & \delta \end{pmatrix} \sigma \left( \begin{pmatrix} \tilde{\mathbf{W}} & \mathbf{0} \\ \mathbf{0} & \tilde{\mathbf{W}} \\ 0 & 1 \\ 0 & -1 \end{pmatrix} \begin{pmatrix} 1 & \frac{1}{2} & \mathbf{0} \\ 1 & -\frac{1}{2} & \mathbf{0} \end{pmatrix} \begin{pmatrix} z \\ x_1 \\ \mathbf{x}_{-1} \end{pmatrix} + \begin{pmatrix} \tilde{\mathbf{b}} \\ \tilde{\mathbf{b}} \\ 0 \\ 0 \end{pmatrix} \right)
$$

$$
= \begin{pmatrix} \frac{\mathbf{a}^T}{2} & \frac{\mathbf{a}^T}{2} & -\delta & \delta \end{pmatrix} \sigma \left( \begin{pmatrix} \tilde{\mathbf{W}} \left( z + \frac{x_1}{2} \right) + \tilde{\mathbf{b}} \\ \tilde{\mathbf{W}} \left( z - \frac{x_1}{2} \right) + \tilde{\mathbf{b}} \\ \left( z - \frac{x_1}{2} \right) \\ -\left( z - \frac{x_1}{2} \right) \end{pmatrix} \right) \tag{29}
$$

$$
= \frac{1}{2} \mathbf{a}^T \sigma \left( \tilde{\mathbf{W}} \left( z + \frac{x_1}{2} \right) + \mathbf{b} \right) + \frac{1}{2} \mathbf{a}^T \sigma \left( \tilde{\mathbf{W}} \left( z - \frac{x_1}{2} \right) + \tilde{\mathbf{b}} \right)
$$

$$
+ \delta \left( \sigma \left( -z + \frac{x_1}{2} - \sigma \left( z - \frac{x_1}{2} \right) \right) \right)
$$

$$
= \frac{1}{2} \left[ \phi_N \left( z + \frac{x_1}{2} \right) - \phi_N \left( z - \frac{x_1}{2} \right) \right] + \delta \left( \frac{x_1}{2} - z \right) = \tilde{h}(z, \mathbf{x}),
$$

where the first line resembles a function implemented by an FNN with width $4N + 2$.

2. $\tilde{h}(z, \mathbf{x})$ approximate $\tilde{g}(z, \mathbf{x})$ well on $(z, \mathbf{x}) \in \left[ -\delta, \frac{1}{2} \right] \times [0, 1]^d$. Since $z + \frac{x_1}{2} \in [0, 1]$ and $z - \frac{x_1}{2} \in [-1, 0]$, from Eq. (28), we have

$$
|\tilde{h}(z, \mathbf{x}) - \tilde{g}(z, \mathbf{x})| = \frac{1}{2} \left[ \phi_N \left( z + \frac{x_1}{2} \right) - \left( z + \frac{x_1}{2} \right)^2 \right] - \frac{1}{2} \left[ \phi_N \left( z - \frac{x_1}{2} \right) - \left( z - \frac{x_1}{2} \right)^2 \right]
$$

$$
\leq \frac{1}{2} \left| \phi_N \left( z + \frac{x_1}{2} \right) - \left( z + \frac{x_1}{2} \right)^2 \right| + \frac{1}{2} \left| \phi_N \left( z - \frac{x_1}{2} \right) - \left( z - \frac{x_1}{2} \right)^2 \right|
$$

$$
\leq \frac{1}{2} \left( \frac{t^2}{4} + \frac{t^2}{4} \right) = \frac{t^2}{4}. \tag{30}
$$

3. The fixed point equation $z = \tilde{h}(z, \mathbf{x})$ is well-posed on $\left[-\delta, \frac{1}{2}\right] \times [0, 1]^d$. for the partial derivative $\frac{\partial \tilde{h}(z, \mathbf{x})}{\partial z}$, we have

$$
\begin{aligned}
\left| \frac{\partial \tilde{h}(z, \mathbf{x})}{\partial z} \right| &= \frac{1}{2} \left( \phi_N' \left( z + \frac{x_1}{2} \right) - \phi_N' \left( z - \frac{x_1}{2} \right) \right) - \delta \\
&\leq \frac{1}{2} \left( \phi_N' \left( z + \frac{x_1}{2} \right) - \phi_N' \left( z + \frac{x_1}{2} - 1 \right) \right) - \delta \\
&\leq 1 - \delta < 1,
\end{aligned}
$$

where the second line holds because $\phi'(x)$ is monotonically increasing and $x_1 < 1$. Therefore, the fixed point equation $z = \tilde{h}(z, \mathbf{x})$ has a unique solution for all $\mathbf{x}$.

4. Note that $\tilde{h}(z, \mathbf{x})$ can be computed by a revised DEQ defined in Eq. (8) with

$$
\mathbf{V} = \begin{pmatrix} \frac{\mathbf{a}^T}{2} & \frac{\mathbf{a}^T}{2} & -\delta & \delta \end{pmatrix}, \quad \mathbf{W} = \begin{pmatrix} \tilde{\mathbf{W}} \\ \tilde{\mathbf{W}} \\ 1 \\ -1 \end{pmatrix}, \quad \mathbf{B} = \mathbf{1}.
$$

And it can be verified that $\| \mathbf{W} \mathbf{V} \|_2 = 1 - t - 2\delta \leq 1$. By Lemma 1, the fixed point of $z = \tilde{h}(z, \mathbf{x})$ can be computed by a DEQ with width $4N + 2$, which we denote by $N_{\text{deq}}(\mathbf{x})$. Further calculations show that the weight of the DEQ is also bounded by $2t$.

**Approximate $g$ using the induced DEQ.** We will bound $\| N_{\text{deq}}(\mathbf{x}) - g(\mathbf{x}) \|_{L^\infty([0,1]^d)}$ using Lemma 2 and Lemma 3. Let $\Omega = \left[ -\delta, \frac{1}{2} \right]$ and assume that $t > 10\delta$. It can be easily verified that both the range of $\tilde{g}(z, \mathbf{x})$ and $\tilde{h}(z, \mathbf{x})$ are in $\Omega$ when $(z, \mathbf{x}) \in \Omega \times [0, 1]^d$.

1. When $x_1 \leq 1 - \frac{t}{2}$, by definition, the Lipschitz constant of $\tilde{g}(\cdot, \mathbf{x})$ is upper bounded by $\max \left| \frac{\partial \tilde{g}(z, \mathbf{x})}{\partial z} \right|$. Leveraging Lemma 2 and Eq.(30), we have

$$
|N_{\text{deq}}(\mathbf{x}) - g(\mathbf{x})| \leq \left| 1 - \frac{\partial \tilde{g}(z, \mathbf{x})}{\partial z} \right|^{-1} |\tilde{h}(z, \mathbf{x}) - \tilde{g}(z, \mathbf{x})| \leq \frac{2}{t} \cdot \frac{t^2}{4} = \frac{t}{2}.
$$

2. When $1 > x_1 > 1 - \frac{t}{2}$, if $z + \frac{x_1}{2} = nt$ for some $\frac{N}{2} - 1 \leq n \leq N$, we have

$$
z - \frac{x_1}{2} = nt - \frac{x_1}{2} \in \left( \left( n - \frac{N}{2} \right) t, \left( n - \frac{N}{2} - 1 \right) t \right).
$$

Note that $\phi_N(x) > x^2$ for all $x \in [0, 1] \backslash t\mathbb{N}$ and $\phi_N(x) = x^2$ for all $x \in [0, 1] \cap t\mathbb{N}$. Thus, when $z = nt - \frac{x_1}{2}$, we have

$$
\tilde{h}(z, \mathbf{x}) < \frac{1}{2} \left( \left( z + \frac{x_1}{2} \right)^2 - \left( z - \frac{x_1}{2} \right)^2 \right) + \delta \left( \frac{x_1}{2} - z \right) = \tilde{g}(z, \mathbf{x}).
$$

For every $T \subset \Omega$ with $|T| \leq t$, there exists $z_g \in T$, such that $z_g = nt - \frac{x_1}{2}$. Thus, from the above analysis, we know that $\tilde{h}(z_g, \mathbf{x}) < \tilde{g}(z_g, \mathbf{x})$.

On the other hand, if $z = \frac{x_1}{2} - kt$ for some $0 \leq k \leq \frac{N}{2} - 1$, we have

$$
z + \frac{x_1}{2} = kt + \frac{x_1}{2} \in \left( \left( -k + \frac{N}{2} - 1 \right) t, \left( -k + \frac{N}{2} \right) t \right).
$$

Similarly, we have

$$
\tilde{h}(z, \mathbf{x}) > \frac{1}{2} \left( \left( z + \frac{x_1}{2} \right)^2 - \left( z - \frac{x_1}{2} \right)^2 \right) + \delta \left( \frac{x_1}{2} - z \right) = \tilde{g}(z, \mathbf{x}).
$$

For every $T \subset \Omega$ with $|T| \leq t$, there exists $z_l \in T$, such that $z_l = \frac{x_1}{2} - kt$. Thus we have $\tilde{h}(z_g, \mathbf{x}) > \tilde{g}(z_g, \mathbf{x})$. From the intermediate value theorem, there exists $z^* \in T$, such that $\tilde{h}(z^*, \mathbf{x}) = \tilde{g}(z^*, \mathbf{x})$. Thus it follows from Lemma 3 immediately that

$$
|N_{\text{deq}}(\mathbf{x}) - g(\mathbf{x})| \leq t.
$$

Additionally, when $x_1 = 1$, by simple calculations, we have $\tilde{h}\left(\frac{1}{2}, \mathbf{x}\right) = \tilde{g}\left(\frac{1}{2}, \mathbf{x}\right) = \frac{1}{2}$, indicating that $N_{\text{deq}}(\mathbf{x}) = g(\mathbf{x}) = \frac{1}{2}$. Combining all the results above, we have

$$|N_{\text{deq}}(\mathbf{x}) - g(\mathbf{x})| \leq |g(\mathbf{x}) - \bar{z}'| \leq t, \quad \mathbf{x} \in [0,1]^d.$$

By choosing $t = \varepsilon$, we finish the proof.

$\square$

### A.3 Proofs in Section 5

We start with the proof of Theorem 3.

*Proof of Theorem 3.* Denote $\{\boldsymbol{\beta}(t)\}$ the process that follows the gradient flow dynamics $\dot{\mathbf{w}}(t) = -\nabla_{\mathbf{w}}\hat{L}(\mathbf{w}(t))$ initialized by $\boldsymbol{\beta}(0) > 0$. Recall that the empirical loss is $\frac{1}{2}\|\mathbf{X}\boldsymbol{\beta} - \mathbf{y}\|_2^2$, then the dynamics of $\{\boldsymbol{\beta}(t)\}$ can be computed as follows:

$$\begin{aligned}
\frac{\mathrm{d}\boldsymbol{\beta}(t)}{\mathrm{d}t} &= \nabla_{\mathbf{w}}\boldsymbol{\beta}(t) \cdot \frac{\mathrm{d}w(t)}{\mathrm{d}t} \\
&= \nabla_{\mathbf{w}}\boldsymbol{\beta}(t) \cdot \left(-\nabla_{\mathbf{w}}\left(\frac{1}{2}\|\mathbf{X}\boldsymbol{\beta}(t) - \mathbf{y}\|_2^2\right)\right) \\
&= \nabla_{\mathbf{w}}\boldsymbol{\beta}(t)^2 \cdot \left(-\nabla_{\boldsymbol{\beta}}\left(\frac{1}{2}\|\mathbf{X}\tilde{\boldsymbol{\beta}}(t) - \mathbf{y}\|_2^2\right)\right) \\
&= -\left(\mathbf{X}^T\mathbf{r}(t)\right) \odot \tilde{\boldsymbol{\beta}}(t)^{\odot 4},
\end{aligned} \tag{31}$$

where $\mathbf{r}(t) = \mathbf{X}\boldsymbol{\beta}(t) - \mathbf{y}$ denotes the residual. For any $t > 0$, it can be verified easily from Eq. (31) that

$$-\frac{1}{3}\boldsymbol{\beta}(t)^{\odot-3} + \frac{1}{3}\boldsymbol{\beta}(0)^{\odot-3} = -\mathbf{X}^T\int_0^t \mathbf{r}(s)\mathrm{d}s. \tag{32}$$

For simplicity, we denote $\mathbf{v}(t) = \int_0^t \mathbf{r}(s)\mathrm{d}s$. Then from Eq. (32), we have

$$\boldsymbol{\beta}(t) = \left(3\mathbf{X}^T\mathbf{v}(t) + \boldsymbol{\beta}(0)^{\odot-3}\right)^{\odot-\frac{1}{3}}. \tag{33}$$

By assumption, $\boldsymbol{\beta}(t)$ converges to some $\boldsymbol{\beta}^\infty \in \mathbb{R}^d$ when $t \to \infty$, thus $\mathbf{v}(t)$ converges to some $\mathbf{v}^\infty := \int_0^\infty \mathbf{r}(s)\mathrm{d}s$. By letting $t \to \infty$ in Eq. (33), we have

$$\boldsymbol{\beta}^\infty = \left(3\mathbf{X}^T\mathbf{v}^\infty + \boldsymbol{\beta}(0)^{\odot-3}\right)^{\odot-\frac{1}{3}}. \tag{34}$$

Next we want to show that $\boldsymbol{\beta}^\infty$ satisfies the KKT condition of the optimization problem in Eq. (12). Given access to the expression of $Q(\boldsymbol{\beta})$, the KKT optimality conditions can be expressed as

$$\mathbf{X}\boldsymbol{\beta}^* = \mathbf{y}, \quad \nabla Q(\boldsymbol{\beta}^*) = \mathbf{X}\mathbf{v},$$

for some $\mathbf{v} \in \mathbb{R}^d$. By the definition of $Q(\boldsymbol{\beta})$, $\nabla Q(\boldsymbol{\beta}^*) = \mathbf{X}^T\mathbf{v}$ is equivalent to

$$\left(\mathbf{X}^T\mathbf{v}\right)_i = (\nabla Q(\boldsymbol{\beta}^*))_i = q'(\beta_i^*) = -(\beta_i^*)^{-3} + \beta_i(0)^{-3}, \quad \forall i.$$

On the other hand, from Eq. (34), it can be verified that

$$-(\beta_i^\infty)^{-3} + \beta_i(0)^{-3} = -3(\mathbf{X}^T\mathbf{v}^\infty)_i - \beta_i(0)^{-3} + \beta_i(0)^{-3} = -3(\mathbf{X}^T\mathbf{v}^\infty)_i, \quad \forall i.$$

Thus it holds that $\nabla Q(\boldsymbol{\beta}^\infty) = -\frac{1}{3}\mathbf{X}\mathbf{v}^\infty$. Combining this with the assumption that $\mathbf{X}\boldsymbol{\beta}^\infty = \mathbf{y}$, we derive that $\boldsymbol{\beta}^\infty$ satisfies the KKT condition. Moreover, it can be observed that $Q(\boldsymbol{\beta})$ is convex, which makes $\boldsymbol{\beta}^\infty$ an optimum of the problem.

$\square$

*Proof of Theorem 4.* **Gradient Flow.** We first show that the distance between $\boldsymbol{\beta}(t)$ and $\hat{\boldsymbol{\beta}}^*$ is bounded. From the dynamic of $\boldsymbol{\beta}(t)$ shown in Eq. (31), we can derive that the gradient flow of $\|\boldsymbol{\beta}(t) - \hat{\boldsymbol{\beta}}^*\|_2^2$ as below:

$$\frac{\mathrm{d}}{\mathrm{d}t}\|\boldsymbol{\beta}(t) - \hat{\boldsymbol{\beta}}^*\|_2^2 = \left(\frac{\mathrm{d}\boldsymbol{\beta}(t)}{\mathrm{d}t}\right)^T (\boldsymbol{\beta}(t) - \hat{\boldsymbol{\beta}}^*) = -\left\|\mathbf{X}(\boldsymbol{\beta}(t) - \hat{\boldsymbol{\beta}}^*) \odot \boldsymbol{\beta}(t)^{\odot 2}\right\|_2^2 \leq 0. \tag{35}$$

Therefore, $\|\boldsymbol{\beta}(t) - \hat{\boldsymbol{\beta}}^*\|_2^2$ is monotonically non-increasing and upper bounded by $\|\boldsymbol{\beta}(0) - \hat{\boldsymbol{\beta}}^*\|_2^2$ for all $t$. By Assumption 1, we have

$$\|\boldsymbol{\beta}(t)\|_\infty \leq \|\boldsymbol{\beta}(t) - \hat{\boldsymbol{\beta}}^*\|_\infty + \|\hat{\boldsymbol{\beta}}^*\|_\infty$$
$$\leq \|\boldsymbol{\beta}(t) - \hat{\boldsymbol{\beta}}^*\|_2 + \|\hat{\boldsymbol{\beta}}^*\|_\infty$$
$$\leq \|\hat{\boldsymbol{\beta}}^*\|_\infty + \|\hat{\boldsymbol{\beta}}^* - \boldsymbol{\beta}(0)\|_2.$$

On the other hand, we also have

$$\|\boldsymbol{\beta}(t)\|_\infty \geq \|\hat{\boldsymbol{\beta}}^*\|_\infty - \|\boldsymbol{\beta}(t) - \hat{\boldsymbol{\beta}}^*\|_2 \geq \|\hat{\boldsymbol{\beta}}^*\|_\infty - \|\hat{\boldsymbol{\beta}}^* - \boldsymbol{\beta}(0)\|_2 \geq c \tag{36}$$

for all $t$. To prove the convergence, we denote $\mathbf{r}(t) = \mathbf{X}\boldsymbol{\beta}(t) - \mathbf{y}$. The gradient flow of $\|\mathbf{r}(t)\|_2^2$ is

$$\frac{\mathrm{d}}{\mathrm{d}t}\|\mathbf{r}(t)\|_2^2 = \left(\frac{\mathrm{d}\boldsymbol{\beta}(t)}{\mathrm{d}t}\right)^T \mathbf{X}^T \mathbf{r}(t) = -(\mathbf{r}(t)^T \mathbf{X}\mathbf{X}^T\mathbf{r}(t)) \odot \boldsymbol{\beta}(t)^{\odot 4}. \tag{37}$$

Combining this with the fact that $\mu_{\min} > 0$ and the lower boundedness of $\|\boldsymbol{\beta}(t)\|_\infty$, we then have

$$\frac{\mathrm{d}}{\mathrm{d}t}\|\mathbf{r}(t)\|_2^2 \leq -c^4 \mu_{\min}\|\mathbf{r}(t)\|_2^2,$$

which proves the convergence of gradient flow. Thus $\boldsymbol{\beta}(t)$ converges to an optima $\boldsymbol{\beta}_f^\infty$. By letting $t \to \infty$ and using the lower boundedness of $\|\boldsymbol{\beta}(t)\|_\infty$, we know that $\|\boldsymbol{\beta}_f^\infty\|_\infty \geq c$.

**Gradient Descent.** The proof of gradient descent follows from a similar strategy. We first give an explicit expression of the update on $\boldsymbol{\beta}^k$. In the following we denote $\mathbf{r}^k = \mathbf{X}\boldsymbol{\beta}^k - \mathbf{y}$. Recall that the gradient descent iterate on $w_i$ is

$$w_i^{k+1} = w_i^k - \eta \frac{1}{(1 - w_i)^2} \mathbf{x}_i \mathbf{r}^k,$$

where $\mathbf{x}_i$ denotes the $i$-th column of $\mathbf{X}$. Then by the definition of $\boldsymbol{\beta}$, we have

$$\beta_i^{k+1} = \frac{1}{1 - w_i^{k+1}} = \left(\frac{1}{1 - w_i^k} - \frac{1}{1 - w_i^k + \eta\beta_i^2 \mathbf{x}_i^T \mathbf{r}^k}\right) \frac{1 - w_i^k}{\eta\beta_i^2 \mathbf{x}_i^T \mathbf{r}^k}$$
$$= \frac{\beta_i^k - \beta_i^{k+1}}{\eta(\beta_i^k)^3 \mathbf{x}_i^T \mathbf{r}^k},$$

Equivalently, the update of $\boldsymbol{\beta}$ can be expressed as

$$\boldsymbol{\beta}^{k+1} = \boldsymbol{\beta}^k - \eta\mathbf{X}^T\mathbf{r}^k \odot \mathbf{u}^k, \quad \mathbf{u}^k := \left(\boldsymbol{\beta}^k\right)^{\odot 3} \odot \boldsymbol{\beta}^{k+1}. \tag{38}$$

We now show that with an appropriate choice of $\eta$, the distance between $\boldsymbol{\beta}^k$ and $\hat{\boldsymbol{\beta}}^*$ is bounded. By Eq. (38), we have

$$\|\boldsymbol{\beta}^{k+1} - \hat{\boldsymbol{\beta}}^*\|_2^2 - \|\boldsymbol{\beta}^k - \hat{\boldsymbol{\beta}}^*\|_2^2 = \|\boldsymbol{\beta}^{k+1} - \boldsymbol{\beta}^k\|_2^2 - 2(\boldsymbol{\beta}^k - \hat{\boldsymbol{\beta}}^*)^T(\boldsymbol{\beta}^{k+1} - \boldsymbol{\beta}^k)$$
$$= \eta^2 \left\|\mathbf{X}^T\mathbf{r}^k \odot \mathbf{u}^k\right\|_2^2 - 2\eta \left\|\mathbf{r}^k \odot (\mathbf{u}^k)^{\odot\frac{1}{2}}\right\|_2^2$$
$$\leq \mu_{\max}\eta^2 \sum_{i=1}^n (r_i^k u_i^k)^2 - 2\eta \sum_{i=1}^n (r_i^k)^2 u_i^k \tag{39}$$
$$= \sum_{i=1}^n \left(\mu_{\max}\eta^2 (u_i^k)^2 - 2\eta u_i^k\right)(r_i^k)^2.$$

Assume $\boldsymbol{\beta}^k > 0$ for all $k$ so that $u_i^k > 0$ for all $i$. Now we set $\eta < \frac{1}{C\mu_{\max}}$. With these conditions, it holds for each $i$ that

$$\mu_{\max}\eta^2 (u_i^k)^2 - 2\eta u_i^k \leq 0.$$

Combining this with Eq. (39), we have $\|\boldsymbol{\beta}^{k+1} - \hat{\boldsymbol{\beta}}^*\|_2^2 \leq \|\boldsymbol{\beta}^k - \hat{\boldsymbol{\beta}}^*\|_2^2$. Similar to Eq. (36), by Assumption 1, it can be shown that $\|\boldsymbol{\beta}^k\|_\infty \geq c > 0$ for all $k$. We then turn to analyze the update of $\|\mathbf{r}^k\|_2$. Note that the square loss function is $\mu_{\max}$-smooth w.r.t. $\boldsymbol{\beta}$, thus we have

$$\|\mathbf{r}^{k+1}\|_2^2 \leq \|\mathbf{r}^k\|_2^2 + 2(\mathbf{r}^k)^T\mathbf{X}(\boldsymbol{\beta}^{k+1} - \boldsymbol{\beta}^k) + \mu_{\max}\|\boldsymbol{\beta}^{k+1} - \boldsymbol{\beta}^k\|_2^2.$$

Substituting the update of $\boldsymbol{\beta}^k$ in Eq. (38) into the above equation, we have

$$\|\mathbf{r}^{k+1}\|_2^2 \le \|\mathbf{r}^k\|_2^2 - 2\eta(\mathbf{r}^k)^T \mathbf{X}\left(\mathbf{X}^T \mathbf{r}^k \odot (\boldsymbol{\beta}^{k+1})^3 \odot \boldsymbol{\beta}^k\right) + \eta^2 \mu_{\max}\left\|\mathbf{X}^T \mathbf{r}^k \odot (\boldsymbol{\beta}^{k+1})^3 \odot \boldsymbol{\beta}^k\right\|_2^2$$

$$= \|\mathbf{r}^k\|_2^2 - 2\eta \sum_{i=1}^{n}(l_i^k)^2 u_i^k + \eta^2 \mu_{\max} \sum_{i=1}^{n}(l_i^k u_i^k)^2,$$

(40)

where we denote $\mathbf{X}^T \mathbf{r}^k = \mathbf{l}^k$ for simplicity. For every fixed $l_i^k$, the quadratic function $f(u_i^k) = -2\eta(l_i^k)^2 u_i^k + \eta^2 \mu_{\max}(l_i^k u_i^k)^2$ attains its minima at $u_i^k = \frac{1}{\eta \mu_{\max}} > C$, from which we know that $f(u_i^k)$ is monotonically decreasing for $u_i^k < C$. Hence, by the fact that $u_i^k > c$, it holds that

$$-2\eta(l_i^k)^2 u_i^k + \eta^2 \mu_{\max}(l_i^k u_i^k)^2 \le (-2\eta c + \eta^2 \mu_{\max} c^2)(l_i^k)^2 \le 0, \quad \forall 1 \le i \le n. \qquad (41)$$

Note that $\sum_{i=1}^{n}(l_i^k)^2 = \|\mathbf{X}^T \mathbf{r}^k\|_2^2 \le \mu_{\max}\|\mathbf{r}^k\|_2^2$. Leveraging this and Eq. (40) and Eq. (41), we have

$$\|\mathbf{r}^{k+1}\|_2^2 \le (1 - (-2\eta c + \eta^2 \mu_{\max} c^2))\|\mathbf{r}^k\|_2^2.$$

Moreover, to ensure $\boldsymbol{\beta}_i^1 = \frac{\boldsymbol{\beta}_i^0}{1+\eta(\boldsymbol{\beta}_i^0)^3 \mathbf{x}_i^T \mathbf{r}^0} \ge 0$ for all $i$, we choose

$$\eta \le \frac{1}{\|\boldsymbol{\beta}^0\|_\infty^3 \mathbf{x}_i^T \mathbf{r}^0} \le \frac{1}{C^3\|\mathbf{X}\|_2\|\mathbf{r}^0\|_2} \le \frac{1}{C^4 \mu_{\max}\|\mathbf{r}^0\|_2}.$$

With this choice of $\eta$, we can prove by induction that the assumption $\boldsymbol{\beta}^k > 0$ holds for all $k$. Indeed, $k = 0$ follows immediately from assumption. If $\boldsymbol{\beta}^t > 0$ holds, then from update of $\boldsymbol{\beta}^{t+1}$, we have

$$\boldsymbol{\beta}_i^{t+1} = \frac{\boldsymbol{\beta}_i^t}{1 + \eta(\boldsymbol{\beta}_i^t)^3 \mathbf{x}_i^T \mathbf{r}^t} \ge \frac{\boldsymbol{\beta}_i^t}{1 + \eta(\boldsymbol{\beta}_i^t)^3 \|\mathbf{x}_i^T\|_2 \|\mathbf{r}^t\|_2} \ge \frac{\boldsymbol{\beta}_i^t}{1 + \eta C^3 \|\mathbf{X}\|_2 \|\mathbf{r}^0\|_2} \ge 0.$$

Thus the induction holds. Finally, we set $\eta = \min\left\{\frac{2}{C^4 \mu_{\max}}, \frac{1}{C^4 \mu_{\max}\|\mathbf{r}^0\|_2}\right\}$, then $\boldsymbol{\beta}^k$ converges to an optimal $\boldsymbol{\beta}_d^\infty$. It follows from $\|\boldsymbol{\beta}^k\|_\infty \ge c$ that $\|\boldsymbol{\beta}_d^\infty\|_\infty \ge c$. We finish the proof. $\qquad\square$

Next, we move to prove Proposition 2. For completeness, we formally introduce the definition of GOTU in [36] as below.

**Definition 1** (Generalization on the Unseen, [36]). *Let $\ell : \mathbb{R} \times \mathbb{R} \to \mathbb{R}$ be a loss function and $\mathcal{S}$ be a given sample space. During training, part of $\mathcal{S}$ is not sampled, which we call the unseen domain $\mathcal{U}$, while in testing, we sample from the full set $\mathcal{S}$. Let $f$ be the target function and $\tilde{f}_{\mathcal{S}\backslash\mathcal{U}}$ the function learned by a learning algorithm on $\mathcal{S}\backslash\mathcal{U}$. The generalization on the unseen for an algorithm $\tilde{f}$ and target function $f$ is defined as*

$$GOTU(f, \tilde{f}, \mathcal{U}) = \mathbb{E}_{X \sim_U \mathcal{U}}[\ell(\tilde{f}_{\mathcal{S}\backslash\mathcal{U}}(X), f(X))],$$

*where $\sim_U \mathcal{U}$ refers to uniform sampling from $\mathcal{U}$.*

*Proof of Proposition 2.* We first give an explicit expression of the expected loss and gradient flow dynamics. Denote

$$\tilde{f}_\beta(\mathbf{x}) = \sum_{i=1}^{d}\beta_i x_i + b = f(\mathbf{w}, \mathbf{x}) = \sum_{i=1}^{d}\frac{1}{1-w_i}x_i + b.$$

By definition, the half $\ell_2$ loss on any sample $\mathbf{x}$ is

$$\ell(\tilde{f}_\beta(\mathbf{x}), f(\mathbf{x})) = \frac{1}{2}(\tilde{f}_\beta(\mathbf{x}) - f(\mathbf{x})) = \frac{1}{2}\left(b - \hat{f}(\emptyset) + \sum_{i=1}^{d}\left(\beta_i - \hat{f}(\{i\})\right)x_i\right)^2$$

Denote the distribution on the training set by $U_{-k}^{d-1}$. Note that $\{1, x_1, \cdots, x_d\}$ are orthogonal in the Hilbert space $\mathcal{S} = \{\pm 1\}^d$ equipped with the inner product $\langle g, h \rangle = \mathbb{E}_{\mathbf{x} \sim_U \{\pm 1\}^d}[g(\mathbf{x})h(\mathbf{x})]$. Denote

the distribution on the training samples by $U_{-k}^{d-1}$. By using Parseval's Theorem, the expected loss on the training set can be expressed as:

$$\mathbb{E}_{U_{-k}^{d-1}}[\ell(\tilde{f}_\beta(\mathbf{x}), f(\mathbf{x}))] = \frac{1}{2}\mathbb{E}_{U_{-k}^{d-1}}\left[\left(b - \hat{f}(\emptyset) + \sum_{i=1}^{d}\left(\beta_i - \hat{f}(\{i\})\right)x_i\right)^2\right]$$

$$= \frac{1}{2}\left(b - \hat{f}(\emptyset) + \beta_k - \hat{f}(\{k\})\right)^2 + \frac{1}{2}\sum_{i\neq k}^{d}(\beta_i - \hat{f}(\{i\}))^2.$$

Then we can derive the gradient flow for $\beta_i$ and $b$ as below

$$\frac{\mathrm{d}b(t)}{\mathrm{d}t} = -(b(t) - \hat{f}(\emptyset) + \beta_k(t) - \hat{f}(\{k\})),$$

$$\frac{\mathrm{d}\beta_k(t)}{\mathrm{d}t} = -(b(t) - \hat{f}(\emptyset) + \beta_k(t) - \hat{f}(\{k\}))\beta_k(t)^4,$$

$$\frac{\mathrm{d}\beta_i(t)}{\mathrm{d}t} = -(\beta_i(t) - \hat{f}(\{i\}))\beta_i(t)^4, \quad \forall i \neq k.$$

For simplicity, denote $B = \hat{f}(\emptyset) + \hat{f}(\{k\})$. Using the above, we have

$$\frac{\mathrm{d}}{\mathrm{d}t}(b(t) + \beta_k(t) - B)^2 = -2(b(t) + \beta_k(t) - B)^2(1 + \beta_k(t)^4),$$

$$\frac{\mathrm{d}}{\mathrm{d}t}(\beta_i(t) - \hat{f}(\{i\}))^2 = -2(\beta_i(t) - \hat{f}(\{i\}))^2\beta_i(t)^4, \tag{42}$$

which shows that $|b(t) + \beta_k(t) - B|^2$ and $|\beta_i(t) - \hat{f}(\{i\})|^2$ is monotonically nonincreasing. Since $\beta_i(0)$ and $\hat{f}(\{i\})$ are greater than 0, from the monotonicity we know that $\beta_i(t) > 0$ for all $t$. Therefore, the convergence of gradient flow follows from Eq. (42) that both $|b(t) + \beta_k(t) - B|^2$ and $|\beta_i(t) - \hat{f}(\{i\})|^2$ decrease linearly.

Denote the limit of $\beta_i(t)$ and $b(t)$ by $\beta_i^\infty$ and $b^\infty$, respectively. We now turn to estimate the GOTU error.

1. When $B > 1$, it holds that $b(0) + \beta_k(0) - B < 0$, thus $b(t)$ and $\beta_k(t)$ is monotonically increasing. Using the fact that $\beta_k(t) > \beta_k(0) = 1$, we know that

$$\frac{\mathrm{d}}{\mathrm{d}t}(\beta_k(t) - b(t)) = -2(b(t) + \beta_k(t) - B)^2(\beta_k(t)^4 - 1) < 0.$$

Combing this with $\beta_k^\infty + b^\infty = B$, it can be verified that $\beta_k^\infty \geq \frac{B+1}{2}$. Then by definition and Parseval's theorem, the GOTU loss is

$$GOTU(f, \tilde{f}_\beta, \{x_k = -1\}) = \left(b^\infty - \hat{f}(\emptyset) - \beta_k^\infty + \hat{f}(\{k\})\right)^2 + \sum_{i\neq k}^{d}(\beta_i^\infty - \hat{f}(\{i\}))^2$$

$$= 4(\hat{f}(\{k\}) - \beta_k^\infty)^2,$$

where we use the convergence of the flow in the second line. Leveraging the bound of $\beta_k^\infty$, we derive that

$$4(\hat{f}(\{k\}) - \beta_k^\infty)^2 \leq 4\left(\hat{f}(\{k\}) - \frac{B+1}{2}\right)^2. \tag{43}$$

By the assumption that $\hat{f}(\emptyset) < 2\hat{f}(\{k\})$, we have $\frac{B+1}{2} < \frac{3\hat{f}(\{k\})+1}{2} < 2\hat{f}(\{k\})$. Leveraging this in Eq. (43), we know that

$$GOTU(f, \tilde{f}_\beta, \{x_k = -1\}) \leq (\hat{f}(\{k\}) + 1)^2. \tag{44}$$

2. When $B < 1$, similar to the proof of Theorem 3, we have from the dynamic of $\beta_k(t)$ that

$$\beta_k(t)^{-3} - 1 = 3\int_0^t (b(s) + \beta_k(t) - B)\mathrm{d}s \leq 3(1 - B),$$

where we use the monotonicity of $b(s) + \beta_k(t) - B$ and the convergence of the flow. Therefore, it holds that $\beta_k^\infty \geq (3(1 - B) + 1)^{-\frac{1}{3}}$. We can bound the GOTU error as

$$4(\hat{f}(\{k\}) - \beta_k^\infty)^2 \leq 4(\hat{f}(\{k\}) - (3(1 - B) + 1)^{-\frac{1}{3}})^2. \tag{45}$$

By using the assumption that $\hat{f}(\emptyset) > -2\hat{f}(\{k\})$, Eq. (45) gives

$$GOTU(f, \tilde{f}_\beta, \{x_k = -1\}) \leq 4 \left( \hat{f}(\{k\}) - \left(4 + 3\hat{f}(\{k\})\right)^{-\frac{1}{3}} \right)^2. \tag{46}$$

Then the proposition follows from Eq. (44) and (46).

$\square$

## A.4 Discussion on Comparison Fairness

In this section, we provide a detailed discussion on the fairness of the comparison in the separation results between the expressivity on DEQs and FNNs.

Concerns may arise regarding the fairness of comparing finite-depth FNNs to DEQs, which resemble infinite-depth weight-tied FNNs. However, as stated in Section 4, our separation results are mainly stated from the actual size of the network (Theorem 2 also considers the parameter magnitude). Characterizing expressivity in term of the depth and width of the network is a common approach as these dimensions can be manipulated in practice. From a theoretical perspective, one primary goal for the comparison is to investigate how the DEQ architecture influences the model capacity with the same number of parameters as FNN whose weights are not tied. This analysis provides intuitions for understanding the model bias of DEQ. Note that one conceptual understanding here is *no free lunch*. Our separation results provide concrete examples of functions that DEQs can approximate more *parameter-efficiently*. Thus, we believe that it is reasonable to compare the expressivity from the actual size as well as the number of parameters of the networks.

Another concern is whether our separation remains valid when considering the actual memory and computational costs associated with training DEQs. If a DEQ needs exponentially more computational cost than FNN, it could be justifiable to compare a DEQ to an FNN of exponentially larger depth. Nevertheless, considering these issues is beyond the scope of this paper since they are also determined by the forward and backward algorithms used. For completeness, we provide some comparisons based on the computational cost of the forward propagation of DEQs as follows.

For the DEQ in Theorem 1, we claim that it needs one step to converge using fixed-point iteration. This is achieved by initializing iteration point $\mathbf{z}^0$ according to Eq. (16) (While it may seem tricky to initialize $\mathbf{z}^0$ as the fixed point, the logic here is that we first observe such convergence under this initialization that we claim this $\mathbf{z}^0$ to be the fixed point). Then from the definition of $\mathbf{W}, \mathbf{U}$ and $\mathbf{b}$, each entry $t$ of $\mathbf{z}^1$ can be calculated as

$$z_t^1 = \sigma \left( \sum_{i=1}^{t-1} w_{ti} z_i^0 + u_{t1} x_1 - b_t \right) = \sigma \left( -\sum_{i=1}^{t-1} 2^{t-i+1} z_i^0 + 2^t x_1 - 1 \right) = z_t^0,$$

showing that it converges in one iteration. Additionally, if we set $\mathbf{z}^0 = \mathbf{0}$, which may be considered less ad hoc, and denote the fixed point by $\mathbf{z}^*$. Then using the lower-triangularity of $\mathbf{W}$, we can show by induction that $\mathbf{z}_t^k = \mathbf{z}_t^*$ for all $1 \leq t \leq k$. Thus, the convergence is achieved in at most $m$ iterations, which still remains far from exponential.

For the DEQ in Theorem 2, we claim that it needs $\log_2(\varepsilon^{-1}) + \log_2(b - a)$ iterations to achieve an $O(\varepsilon)$-solution using bisection method. Given the DEQ, we can derive a revised DEQ, i.e., $\mathbf{z} = \mathbf{V}\sigma(\mathbf{W}z + \mathbf{U}x + \mathbf{b})$ based on our Lemma 1 (One can inversely derive the revised DEQ through rank-one decomposition, and we admit any version of it). Then for the function $z - \mathbf{V}\sigma(\mathbf{W}z + \mathbf{U}x + \mathbf{b})$, we choose some $[a^0, b^0]$ as the initial interval such that this function has opposite signs on $a^0$ and $b^0$. Then after $k$ iterations the solution $z^*$ will lie within an interval $[a^k, b^k]$. From the definition of the algorithm, we know that $b^{i+1} - a^{i+1} = 2^{-1}(b^i - a^i)$, leading to $b^k - a^k \leq 2^{-k}(b^0 - a^0)$. To achieve an $O(\varepsilon)$-solution, i.e., $2^{-k}(b^0 - a^0) \leq \varepsilon$, we can require $k \geq \log_2 \varepsilon^{-1} + \log_2(b^0 - a^0)$, which proves our claim.

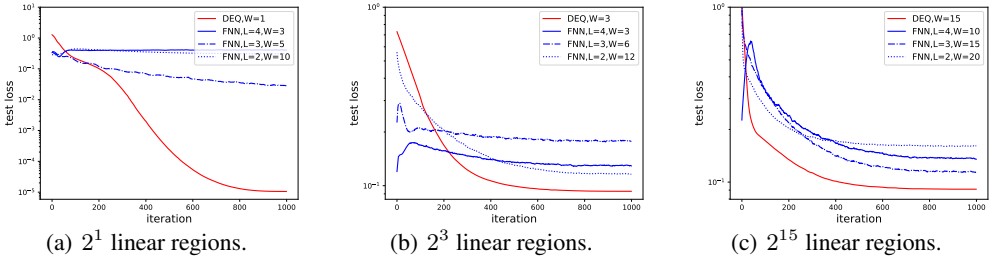

(a) $2^1$ linear regions.     (b) $2^3$ linear regions.     (c) $2^{15}$ linear regions.

Figure 3: Test loss of FNN and DEQ trained on sawtooth functions with $2^1$, $2^3$, $2^{15}$ linear regions.

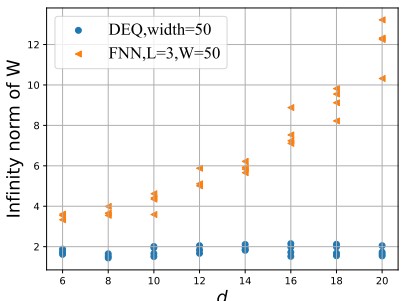

Figure 4: Results on steep target functions applying DEQ and FNN with comparable performance on loss. We plot the infinity norm of the weights of DEQ and FNN with $d = 6, 8, 10, 12, 14, 16, 18, 20$.

For the memory cost, DEQ's memory usage is comparable to that of constant-depth FNN. Since the DEQs we consider in our theorems have fewer parameters, they are more advantageous than the corresponding FNNs in memory consumption. Consequently, our comparisons are likely to remain valid even when actual memory and computational costs are taken into account.

# B   Experiment Details

## B.1   Supplementary Experiments on Separation Results

**More experiments on piecewise functions**. We first experiment on other sawtooth functions with varying number of linear regions in the setting of Figure 3 to further validate our theoretical results. Figure 3 presents the test loss for sawtooth functions with $2^1$, $2^3$ and $2^{15}$ linear regions. For all experiments, we execute our programs on Nvidia GTX 1660 and all the programs occupy less than 10M memory and run for less than 2 minutes. Consistent with our results in Section 6, we observe that DEQ outperforms FNN with similar size of network on every sawtooth function in our experiment and the test loss of DEQ converges closer to zero loss.

Additionally, we conduct an experiment to investigate the performance of DEQ and FNN on sawtooth function with similar actual computational cost to further strengthen the fairness as discussed in Appendix A.4. Following the standard setting, all models in our experiment are trained using $\ell_2$ loss with AdamW optimizer [44], with a learning rate of 5e-4, weight decay of 1e-4 and a cosine annealing scheduler for 1000 iterations. Table 1 indicates that when training on sawtooth function under similar computational cost, DEQ also performs better than FNN.

**More experiments on steep target functions.** To verify the statements of the magnitude of weights in Theorem 2, we add an experiment to examine the infinity norm of the weights of DEQ and FNN when training on steep target functions with varying $d$, as shown in Figure 4. In this experiment, we train a 3-layer FNN and a 50-width DEQ model using the $\ell_2$ loss. Following standard settings, we employ a mini-batch SGD optimizer with a learning rate of $0.005$, weight decay of 1e-4 and a cosine annealing scheduler for all models. The infinity norm of the weights of DEQ remains approximately

| Model | Parameter number | layer | hidden dimension | Average FLOPs/Iter | Loss |
|-------|------------------|-------|------------------|-------------------|------|
| DEQ | 251 | - | - | 3.44e5 | $0.078 \pm 0.004$ |
| FNN | 2401 | 4 | 30 | 3.18e5 | $0.094 \pm 0.006$ |
| FNN | 2281 | 3 | 40 | 3.03e5 | $0.082 \pm 0.004$ |
| FNN | 2801 | 2 | 200 | 3.84e5 | $0.085 \pm 0.007$ |

Table 1: Training settings of experiment. We apply DEQ with width-10 and FNN trained on sawtooth target function with $2^5$ linear regions while keeping FLOPs per iteration similar.

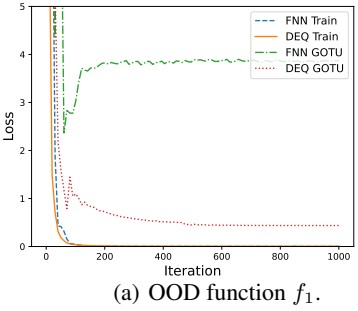

(a) OOD function $f_1$.

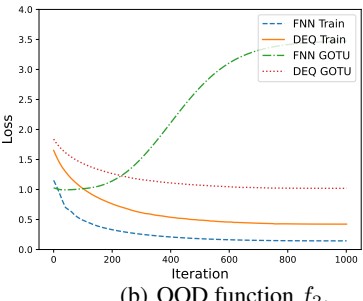

(b) OOD function $f_2$.

Figure 5: Train and test loss of DEQ and FNN trained on OOD tasks $f_1$ and $f_2$.

constant when $d$ increases, while the infinity norm of FNN rises with $d$, verifying the theoretical claims that DEQ with polynomially bounded size and magnitude can efficiently approximate certain steep functions in infinity norm while FNN cannot.

## B.2 Experiments on the Bias on Learning Dynamics

**More experiments on Out-of-Distribution tasks.** We conduct more out-of-distribution (OOD) experiments on the following 2 functions and unseen domains. The first function is a higher-dimensional extension of Eq. (14) which is a form of mean function. The second function is the majority function for 3 bits with a maximum degree of 3. The mathematical expressions of these functions are presented below.

$$f_1(x) = 1.25 * x_0 + 1.25 * x_1 + \cdots + 1.25 * x_{20}, \quad \mathcal{U} = \{\mathbf{x} \in \{\pm 1\}^{10} : x_1 = -1\},$$

$$f_2(x) = \frac{1}{2}(x_0 + x_1 + x_2 - x_1 x_2 x_3), \quad \mathcal{U} = \{\mathbf{x} \in \{\pm 1\}^{10} : x_0 x_1 = -1\}.$$

For all experiments, we generate all binary sequences in $\mathcal{U}^c = \{\pm 1\}^d \backslash \mathcal{U}$ for training. Figure 5(a) shows that the GOTU error does not increase significantly compared to Figure 1 where the ambient dimension is 10. In consistency with our results in Section 6, we can learn from Figure 5(a) that when learning a linear boolean function on population loss on DEQ, the training loss converges to zero and the generalization error on the unseen is bounded. As shown in Figure 5(b), when learning the unlinear boolean function, DEQ can also achieve nearly zero train loss with smaller GOTU error compared with FNN.

**Experiment on salinecy map**. We further provide a visualization experiment to investigate whether the 'dense' bias property still holds for more general DEQs, particularly those with inner structures resembling ResNet or Transformer architectures. Specifically, we conduct an experiment using Grad-CAM [45] to generate the saliency map of Multiscale DEQ (MDEQ [3]), which is a variant of ResNet-DEQ and compare it with that of ResNet-50 trained for image classification on ImageNet [46]. The saliency map highlights the regions that are crucial for the model's prediction, which can be regarded as the features. It is shown in Figure 6 that MDEQ allocates attention to more features such as the fences and trees in the background, indicating that MDEQ may generate dense features. The intuitive experiment suggests that our bias results may be applicable to a wider array of DEQs with general architectures.

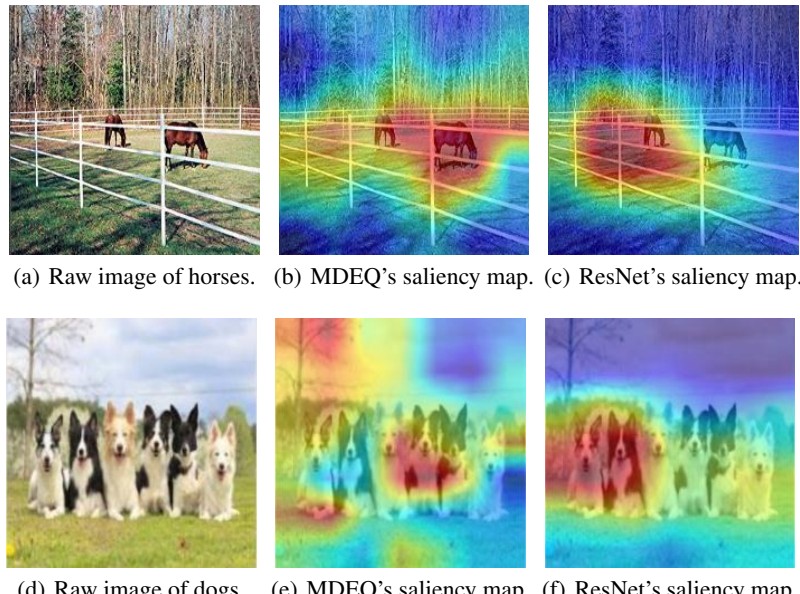

(a) Raw image of horses. (b) MDEQ's saliency map. (c) ResNet's saliency map.

(d) Raw image of dogs. (e) MDEQ's saliency map. (f) ResNet's saliency map.

Figure 6: The saliency map of Multiscale DEQ (MDEQ) and ResNet.

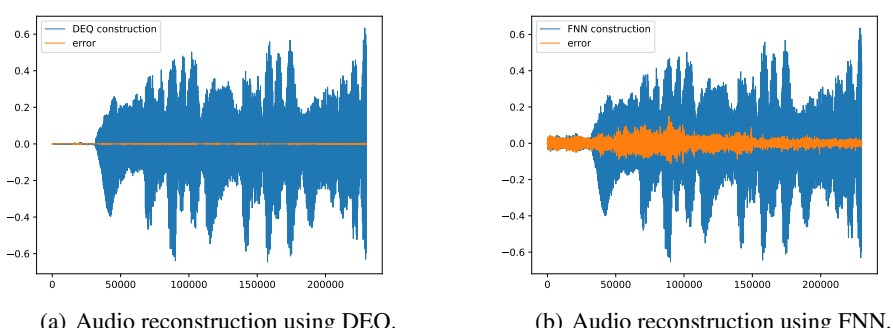

(a) Audio reconstruction using DEQ.  (b) Audio reconstruction using FNN.

Figure 7: The reconstruction results with DEQ and FNN and the error computed by subtracting the original signal.

**Experiment on audio representation**. Inspired by the overall studies, we conduct an experiment on a real task of audio representation to verify the potential advantage of DEQ in learning functions with high-frequency component. We utilize the setting of experiments in [47], where the very-high-frequency audio signals were represented using a conventional explicit network and an $(\text{implicit})^2$ network, which is a variant of DEQ employing a neural block with three layers and specific activation functions such as $\sin(x)$ Although Huang et al. [47] show that $(\text{implicit})^2$ network outperforms conventional explicit networks in audio representation [47], revealing the advantage of DEQ to an extend, it is unclear whether the superiority of the $(\text{implicit})^2$ network is attributed solely to the carefully-designed block. In contrast, we apply DEQ and FNN in their basic forms to represent the audio signal in our experiment to further explore the potential advantages of DEQ in real scenarios.

For our audio representation task, we aim to train a function that can reconstruct an given audio signal over a period of time. The spectrum of an audio signal contains many high-frequency components, making it difficult to reconstruct the audio signal for a not-so-large FNN.

Following the setting in [47], we train the models to fit a 7-second music piece. We set the width of DEQ to 20, the layer of FNN to 3 and the hidden dimension of FNN to 20. This setting enables the model to exactly fit the audio signal based on our experiments.

In Figure 7, we show the reconstruction results with DEQ and FNN and the error computed by subtracting the original signal. We observe that DEQ outperforms FNN with a noticeable error, verifying the advantages of DEQ in representing high-frequency components.

