# OpenReview forum: "Separation and Bias of Deep Equilibrium Models on Expressivity and Learning Dynamics"
_NeurIPS.cc/2024/Conference — NeurIPS 2024 poster_

### Official Review · Reviewer_QS3W · 2024-07-05

**Soundness:** 3
**Presentation:** 3
**Contribution:** 3
**Rating:** 6
**Confidence:** 4

**Summary:**

This paper offers a comparative analysis of DEQ and FNN, examining their differences in terms of structure and performance. By investigating the learning dynamics, the authors provide insights into the implicit bias of DEQ. Their theoretical findings demonstrate the advantages of DEQ over FNN in specific scenarios and quantify key learning properties unique to DEQ.

**Strengths:**

- This paper is well-written and has a clear logic.

- There is relatively little theoretical research on DEQ, and this paper serves as a valuable contribution.

- From the perspective of separation, the authors discuss the relationship between the width of DEQ and the number of layers in FNN. They also use gradient flow and gradient descent to explore DEQ's implicit bias, clearly explaining the properties of DEQ.

**Weaknesses:**

Factors influencing DEQ's separation and bias need further discussion. For instance, is DEQ's implicit bias caused by initialization, the use of implicit differentiation in solving DEQ, or different gradient descent methods? The reasons behind its differences from FNN should be explained in detail.

Besides, the following relevant papers should be cited and discussed:
[1] Deep equilibrium networks are sensitive to initialization statistics, icml 2022
[2] GEQ: Gaussian Kernel Inspired Equilibrium Models, neurips 2023
[3] Wide Neural Networks as Gaussian Processes: Lessons from Deep Equilibrium Models, neurips 2023
[4] Deep Equilibrium Models are Almost Equivalent to Not-so-deep Explicit Models for High-dimensional Gaussian Mixtures, icml 2024

**Questions:**

- Can the implicit bias of DEQ be understood from a lazy training perspective?
- In Theorem 4.1, how is $\alpha$ defined? For example, if $\alpha$ is large and close to 1, then $m^{1-\alpha}$ would be particularly small, implying that $N_f$ would have almost no width.
- Will the conclusion differ under different circumstances of DEQ? For instance, if the inner structure is a ResNet or Transformer, will the conclusion change? Could the authors provide more experiments testing different backbones?
- Additionally, does the conclusion hold for various downstream tasks of DEQ, including classification, regression, or other tasks?

---

> ### Author Rebuttal · Authors · 2024-08-07
>
> Thank you for your careful review and thoughtful questions and comments. Please see our response below.
>
> > Factors influencing DEQ's separation and bias need further discussion. For instance, is DEQ's implicit bias caused by initialization, the use of implicit differentiation in solving DEQ, or different gradient descent methods? The reasons behind its differences from FNN should be explained in detail.
>
> Thanks for your suggestion. Our result of the bias of DEQ is indeed influenced by the optimization algorithms, the initialization, and possibly by the use of implicit differentiation (Although we do not consider the use of implicit differentiation in this paper). It is interesting to study the bias caused by other gradient algorithms such as Adam. We believe that different gradient algorithms may bring different biases on DEQ as it has been shown empirically that different gradient algorithms cause different implicit biases on FNNs in many works. The initialization is also an important factor and our Theorems also need assumptions on the initialization. We consider the Gradient Descent algorithm and a usual initialization in this paper since the paper proposes the first analysis for the implicit regularization of DEQ beyond the NTK regime. Different algorithms and initializations will be studied in future works. Regarding reasons why DEQ has different expressive power and bias from FNN, we believe that it is mainly because of the difference in architecture. We will add discussions to elucidate these issues in the final version of the paper.
>
> > Besides, the following relevant papers should be cited and discussed.
>
> Thanks for your valuable feedback. We will include all papers you mentioned in the final version of the paper.
>
> > Can the implicit bias of DEQ be understood from a lazy training perspective?
>
> We think that our result on the bias of DEQ is beyond the lazy training regime because the diagonal linear DEQ we consider is essentially a nonlinear model, albeit with some simplifications. Unlike the lazy training scenario, we do not require the DEQ to be sufficiently wide, nor do we require the updates to stay in a small domain near the initialization. Therefore, our result of the bias of DEQ is beyond the lazy training perspective. Nevertheless, our proofs of the convergence of DEQ indeed utilizes some techniques similar to those of NTK. We show that the simplified DEQ is convex in a relatively large domain under some assumptions of the initialization and we use induction to ensure the updates remain within this domain throughout the training. Thank you for your valuable question and we hope our response address your concerns.
>
> > In Theorem 4.1, how is α defined? For example, if α is large and close to 1, then m1−α would be particularly small, implying that Nf would have almost no width.
>
> We are sorry for the confusion in the statement of Theorem 4. In our proof, we actually assume that the width $k$ is upper bounded by $2^{m^{1-\alpha}-1}$ instead of $2^{m^{1-\alpha}-2}$ (see the equation before line 424 in Appendix) so that $k$ can take value $1$ even when $\alpha$ is large and very close to 1. Although the original statement is not wrong as the inapproximability pertains to a shallower FNN with less expressive power, it does lead to a problem of defining $\alpha$ to avoid FNN having no width. We will address this issue in the final version of the paper. In this case, $\alpha$ is an integer in $(0,1)$, making $2^{m^{1-\alpha}}-1$ sub-exponentially large when $m\to \infty$. Thank you very much for your valuable feedback.
>
> > Will the conclusion differ under different circumstances of DEQ? For instance, if the inner structure is a ResNet or Transformer, will the conclusion change? Could the authors provide more experiments testing different backbones?
>
> It is very interesting to consider the expressivity and bias of DEQ when the inner structure is a ResNet or a Transformer. The expressivity and bias of DEQ depends on the specific inner structure so that the conclusions may change. However, this paper aims to provide the first analysis for the implicit regularization of DEQ beyond NTK regime, so we focus on the fundamental FFN as the inner structure. We will analyze the properties of DEQ with other inner structures in the future. To further investigate this issue, we use Grad-CAM [1] to generate the saliency map of Multiscale DEQ (MDEQ [2], which is a variant of ResNet-DEQ) and compare it with that of ResNet-50 trained for image classification on ImageNet. The saliency map highlights the regions that are crucial for the model’s prediction, which can be regarded as the features. It is shown that MDEQ allocate attention to more features such as the fences and trees in the background, indicating that MDEQ may generate dense features. Please see our attached PDF file in Author Response for details.
> ***
> [1] Grad-CAM: Visual Explanations from Deep Networks via Gradient-Based Localization, ICCV 2017.
>
> [2] Multiscale Deep Equilibrium Models, NeurIPS 2020.
>
> > Additionally, does the conclusion hold for various downstream tasks of DEQ, including classification, regression, or other tasks?
>
> The conclusions for the expressive power of DEQ is not influenced by downstream tasks. However, the bias of DEQ may differ depending on the specific downstream tasks. For example, in classification tasks, the loss function is typically chosen as logistic loss or cross-entropy loss, which may lead to different analyses compared to the $L^2$ loss in regression problems. The implicit bias for linear models in classification problems and regression problems has been studied separately in numerous works. We think that it is interesting to further investigate the bias of DEQ in various tasks such as classification or sequence modeling and we leave them as future directions.

---

### Official Review · Reviewer_j6dr · 2024-07-12

**Soundness:** 2
**Presentation:** 3
**Contribution:** 3
**Rating:** 6
**Confidence:** 4

**Summary:**

- The authors study the expressive power of the ReLU-based deep equilibrium model (DEQ) and the learning dynamics (including implicit bias, convergence, and generalization) of linear DEQ.
- The first separation result shows that there exists a width $m$ ReLU-DEQ that cannot be well-approximated by a ReLU-based fully-connected neural network (FNN) with sub-linear depth and sub-exponential (super-polynomial) width.
- The second separation result shows that a rational function, which can be $\epsilon$-approximated by a RELU-DEQ with $O(\epsilon^{-1})$ width and weights, cannot be well-approximated by a RELU-FNN with constant depth and sub-exponential depth (in dimension) unless its weights are exponentially large (in dimension).
- They also proved a certain implicit bias result for linear DEQ. They claim that the implicit bias implies that GF (or GD) learns a dense feature.
- Under the generalization on the unseen domain (GOTU) setting, they proved that linear DEQ with a bias term achieves a smaller GOTU error than a diagonal linear neural network.
- Overall, they hypothesize that DEQ is beneficial for learning certain high-frequency features.

**Strengths:**

- The paper is well-written and well-organized. If the reader has knowledge about the expressive power of neural networks and the implicit bias of diagonal linear networks, the paper is easy to follow.
- The contribution seems clear if all the proofs are correct. The authors propose novel theoretical results on the properties of DEQ. Also, their result explicitly shows the advantage of DEQ over FNN.

**Weaknesses:**

- W1. It is unclear whether the comparison between DEQ and FNN is fair.
    - Of course, in terms of memory consumption, DEQ is comparable to a constant-depth FNN of the same width. However, it might be unfair to compare them if a DEQ requires a lot more computational cost than FNN during training. For example, suppose a DEQ needs exponentially more computational cost than FNN. Then it might be plausible to compare a DEQ with an FNN of exponentially large depth. Therefore, the authors must provide a comparison in terms of both memory consumption and computational cost (during training) to justify the separation results and claim that DEQ is better than FNN.
- W2. Issues in the proof of Theorem 2(B). (Section 4.3)
    - The proof repeatedly uses an inequality $| \hat{u}(\hat{v}(z)) - \hat{v}(\hat{v}(z)) | \le | \hat{u}(z) - \hat{v}(z) |$. However, I cannot understand why this is true. If it isn’t wrong, the authors should provide proof of this. For the same reason, I am a bit suspicious about the verity of Lemma 2.
- W3. Critical issues in Section 5 (Implicit bias part).
    - If you want to discuss the implicit bias of a given model in Equation 9, the model must be over-parameterized ($N\le d$) at first so that the model can fit all the data points. In other words, if the model is under-parameterized ($N>d$), the model can never fit all the training examples and the training loss cannot even reach zero. However, I cannot find any such discussion. (Please correct me if I am wrong.)
    - You can find that the function $q(x)$ is nonconvex and unbounded below unless $x>0$. However, the solution space (defined as $X\beta = y$) may include a vector $\beta$ with negative components. Then can you guarantee that the function $Q(\beta)$ is convex (and bounded below) over the solution space with ease? For this reason, I strongly believe that the validity of Theorem 3 must be reconsidered.
- W4. The main paper must provide pointers to the proofs of the theorems.
- W5. Minor comments on typos & mistakes
    - Equation (1): “$1\le i \le L-1$”, “$y = W_L z^{L}$”
    - Equation (3): “$A = (0, \cdots, W_L)$”
    - Theorem 1 must include that $m\ge 3$; otherwise, it allows the width (of FNN) less than 1. Also, the function $N_f$ must be defined on $[0, 1]^d$.
    - Proposition 1 must specify $\sigma=$ReLU.
    - Since the paper focuses on linear DEQ from Section 5, the authors must make clear that they consider ReLU-DEQ in Section 4. This must be considered in the explanation in Section 3 (especially in line 112).
    - Lemma 1 must include the size of $\mathbf{b} \in \mathbb{R}^q$.
    - The last sentence of Lemma 2 is a bit awkward because it has two “for all” phrases. You may replace “Then for any $x \in [0,1]^d$,” with “Then,”.
    - Line 197: “$x_1 < 1-\text{poly}(d)^{-1}$”.
    - Lines 212-213: it would be better to mention that $\sigma$ is no longer a ReLU.
    - Line 219: “initialized”
    - Line 221: the function $q(x)$ must have a dependency on the index $i$. For example, write as $Q(\boldsymbol{\beta}) = \sum_{i=1}^d q_i (\beta_i)$.
    - Line 225: “$\tfrac{1}{2} \sum_{i=1}^d \tfrac{1}{\beta_i^2}$”
    - Line 233: a full stop (’.’) is missing.
    - Line 239: “constraint”.
    - Line 248, see the line next to Equation (13): it seems that $X$ (next to the expectation symbol) must be a small, bold letter ($\mathbf{x}$).

I will definitely raise my score if all my concerns are properly addressed.

**Questions:**

- Q1. Can you explain what $f(\emptyset)$ is in Equation 13? Is it an arbitrary number?

**Limitations:**

The main paper does not clearly provide its limitations. However, as provided in the checklist, the main limitation is the simplified models for theoretical analyses. This barrier is difficult to overcome, which is understandable given the paper’s novelty.

---

> ### Author Rebuttal · Authors · 2024-08-04
>
> Thank you very much for your careful review and thoughtful questions and comments. Please see our response below.
>
> > W1. It is unclear whether the comparison between DEQ and FNN is fair. …. Therefore, the authors must provide a comparison in terms of both memory consumption and computational cost (during training) to justify the separation results and claim that DEQ is better than FNN.
>
> We admit that memory consumption and computational cost are vital in determining the model performance in real applications. However, considering these issues is beyond the scope of the expressivity of DEQ since they are also determined by the optimization and forward and backward algorithms used. The main goal for our comparison is to study the bias and potential advantage of DEQ compared to FFN on expressivity with a comparable number of parameters. This analysis provides intuitions for understanding the model bias of DEQ.
>
> However, to address your concerns, we also provide some comparisons based on the computational cost of the forward propagation of DEQs. In our separation result (Theorem 1), the DEQ only need one iteration to converge in the forward propagation using fixed-point iteration, making its computational costs comparable to that of a constant-depth FFN with the same width. The DEQ in Theorem 2 needs $\log_2(\varepsilon^{-1}) + \log_2(b-a)$ iterations to achieve an $O(\varepsilon)$-solution using bisection method starting from an interval $[a,b]$ with constant length, resulting in a cost comparable to that of $O(\log(\varepsilon^{-1}))$-depth FFN with the same width. For the memory cost, DEQ’s memory usage is comparable to that of constant-depth FFN. Since the DEQs we consider in our theorems have fewer parameters, they are more advantageous than the corresponding FNNs in memory consumption. It is interesting to consider the computational and memory cost in a more general setting, and we will explore them in the near future. Additionally, we add an experiment showing that DEQ can achieve better performance with similar FLOPs per iteration compared to various FFNs. Please see our attached PDF file in Author Response for details.
>
>
> > W2. Issues in the proof of Theorem 2(B). (Section 4.3)
>
> We are sorry for the confusion. Lemma 2 should be stated as: “Then for any $\mathbf{x} \in [0,1]^d$, it holds that $$|z_u – z_v| \leq \min \left\\{ (1-L_u)^{-1}, (1-L_v)^{-1} \right\\} \cdot \max_{z\in \Omega}|u(z,\mathbf{x}) – v(z, \mathbf{x})|. $$”
> And the inequality you mention should be $|\hat{u}\circ \hat{v}(z) - \hat{v} \circ \hat{v}(z)| \leq \max_{z\in \Omega}|\hat{u}(z) - \hat{v}(z)|$. This is why we need the range of $u(z,\mathbf{x})$ and $v(z,\mathbf{x})$ to be in $\Omega$ in Lemma 2. This will not influence the proof of Theorem 2 since we construct approximation in the sense of infinity norm to achieve $\lVert \tilde{h}-\tilde{g}\rVert _{\infty} \leq \text{poly}(d)^{-1}$, where $\tilde{h}$ and $\tilde{g}$ correspond to $u$ and $v$ in Lemma 2. We will carefully revise the statement of Lemma 2 and all related statements in the final version of the paper.
>
> > Critical issues in Section 5 (Implicit bias part).
> If you want to discuss the implicit bias of a given model in Equation 9, the model must be over-parameterized (N≤d) at first so that the model can fit all the data points.
>
> We agree that the model must be over-parameterized to fit all data points. In Section 5, we also consider the over-parameterized setting by taking it as a condition or assumption for the theorems. For example, Theorem 3 states, “Suppose the gradient flow converges to some \hat{\bm{beta}} satisfying $\mathbf{X}\hat{\mathbf{\beta}} = \mathbf{y}$ “, which assumes that the model can fit all training examples. We will add clarifying statements at the beginning of 5 to emphasize that we consider over-parameterized DEQ in this Section to avoid any confusion.
>
>
> > Then can you guarantee that the function Q(β) is convex (and bounded below) over the solution space with ease? For this reason, I strongly believe that the validity of Theorem 3 must be reconsidered.
>
> Theorem 3 assumes that the model is initialized as $\beta_i(0)>0$ and the gradient flow converges to a global minimum $\hat{\mathbf{\beta}}$. From our proof, the dynamic of $\mathbf{\beta}(t)$ is $\dot{\mathbf{\beta}}(t) = -(\mathbf{X}^T \mathbf{r}(t)) \odot \mathbf{\beta}(t)^{\odot 4}$. Thus, $\mathbf{\beta} (t)$ will always remain positive for every entry in the training process since the trajectory cannot cross over $0$ by the continuity of gradient flow. In the space of $\mathbf{\beta}>0$, $Q(\mathbf{\beta})$ is convex and has a unique minimum, therefore, we believe that Theorem 3 is valid. As for why Theorem 3 only admits solutions with positive entries, it is because we simplify the model by setting the linear transformation in the last layer of DEQ to be an all-one vector for tractability and simplicity. To consider solutions with negative entries, we can set the corresponding entry (say entry $i$) in the linear transformation to be $-1$, and the corresponding $q_i(x)$ in Theorem 3 will still be $q_i(x) = \frac{1}{2x^2} - \beta_i(0)^{-3}x$ with $ \beta_i(0)<0$, which is convex in the space of $x<0$. We will include a remark and a discussion on these issues in the final version of the paper.
>
> > The main paper must provide pointers to the proofs of the theorems.
>
> Thanks for your valuable suggestions. We will provide pointers to the proofs for all theorems in the final version of the paper.
>
> > Minor comments on typos & mistakes.
>
> Thanks for your careful review. We will revise all the typos and confusions in the final version of the paper.
>
> > Q1. Can you explain what f(∅) is in Equation 13? Is it an arbitrary number?
>
> $\hat{f}(\emptyset)$ is the Fourier coefficient of $f$ on $\emptyset$, which is defined as $\hat{f}(\emptyset) = \mathbb{E}_{\mathbf{x} \sim \\{-1,1\\}^d}[f(\mathbf{x})]$. We will add a statement to clarify it in the final version of the paper.

---

> ### Comment · Reviewer_j6dr · 2024-08-09
> **Remaining Questions**
>
> Thank you for your time and effort in preparing the author's rebuttal. Still, I have some minor remaining questions/comments.
>
> * W1: I am mostly happy with your response. But could you provide some references or simple proofs for your claims about computational costs?
>
> * W2: This response answers my question.
>
> * W3-1: In fact, after posting the review, I found there was already a remark about "overparameterized regime" in the paper (Line 218). Thus, I want to withdraw my statement "I cannot find any such discussion." Nevertheless, it would be meaningful if the authors clarified that they are only interested in the case where $N\le d$ in order to study the implicit bias of diagonal DEQ. Also, putting a comment about "$\mu_{\min}>0$ can hold when $N\le d$ and the data matrix $\boldsymbol{X}$ is of full rank" would strengthen the writing. Moreover, I recommend putting a remark like "the current implicit bias holds for GD... other optimization algorithms may lead to different implicit biases, which we leave as an interesting future work...". I don't think that not studying other algorithms than GD is a weakness of the paper.
>   * By the way, can we even train a usual DEQ with GD? If it isn't, it might be a problem because the paper's implicit bias result can be seen as an artificial result just for writing a paper. Nonetheless, I understand it is difficult to study the implicit bias of the true learning algorithm for DEQ and I don't want to decrease my score.
>
> * W3-2. This might be a stupid question, but could you explain why $\beta(t)$ remains positive (for every coordinate) in more detail?
>
> Thank you again for your reply. After getting responses to these additional questions, I will reconsider my score.  Also, if there are missing details for the response above due to the space limit, please leave them as a comment here, then I'll happily read them all.

---

> > ### Author Response · Authors · 2024-08-10
> >
> > We sincerely thank you for taking time to review our response. We would like to further response to your questions and comments:
> >
> > __Response to W1__:
> > * We provide the proofs of our claims about the computational costs as follows. For the DEQ in Theorem 1 (see Line 405-406 in Appendix A.1), we claim that it needs one step to converge using fixed-point iteration. This is achieved by initializing iteration point $\mathbf{z}^0$ according to Eq. (16) (While it may seem tricky to initialize $\mathbf{z}^0$ as the fixed point, the logic here is that we first observe such convergence under this initialization that we claim this $\mathbf{z}^0$ to be the fixed point). Then from the definition of $\mathbf{W}, \mathbf{U}$ and $\mathbf{b}$, each entry $t$ of $\mathbf{z}^1$ can be calculated as
> > $$z^1_t =\sigma\left(\sum_{i=1}^{t-1} w_{ti}z^0_i + u_{t1} x_1 -b_t \right) =  \sigma\left(-\sum_{i=1}^{t-1} 2^{t-i+1}z^0_i + 2^t x_1 -1\right) = z^0_t,$$
> > showing that it converges in one iteration. Additionally, if we set $\mathbf{z}^0 = \mathbf{0}$, which may be considered less ad hoc, and denote the fixed point by $\mathbf{z}^*$. Then using the lower-triangularity of $\mathbf{W}$, we can show by induction that $\mathbf{z}^k_t = \mathbf{z}^*_t$ for all $1\leq t \leq k$. Thus, the convergence is achieved in at most $m$ iterations, which still remains far from exponential.
> >
> > * For the DEQ in Theorem 2, we claim that it needs $\log_2(\varepsilon^{-1}) + \log_2(b-a)$ iterations to achieve an $O(\varepsilon)$-solution using bisection method. You can refer to Page 29 in [1] for the convergence rate of bisection method. The proof can be stated it as follows: Given the DEQ, we can derive a revised DEQ, i.e., $\mathbf{z} = \mathbf{V}\sigma(\mathbf{W}z +\mathbf{U}\mathbf{x} + \mathbf{b})$ based on our Lemma 1 (See Line 476-477 for the construction. We can inversely derive the revised DEQ through rank-one decomposition, and we admit any version of it). Then for the function $z - \mathbf{V}\sigma(\mathbf{W}z +\mathbf{U}\mathbf{x} + \mathbf{b})$, we choose some $[a^0, b^0]$ as the initial interval such that this function has opposite signs on $a^0$ and $b^0$. Then after $k$ iterations the solution $z^*$ will lie within an interval $[a^k, b^k]$. From the definition of the algorithm, we know that $b^{i+1} – a^{i+1} = 2^{-1} (b^i – a^i)$, leading to $b^k – a^k \leq 2^{-k}(b^0-a^0)$. To achieve an $O(\varepsilon)$-solution, i.e., $2^{-k}(b^0-a^0) \leq \varepsilon $, we can require $k \geq \log_2{\varepsilon^{-1}} + \log_2(b^0-a^0)$, which proves our claim.
> > ***
> > An Introduction to Numerical Analysis, Cambridge University Press, 2003.
> >
> >
> > __Response to W3-1__: Thanks very much to your suggestions. We will add clarifying statements to emphasize that we only consider $N\leq d$ and include the comment and remark you mentioned in Section 5 and the conclusion section in the final version of the paper.
> >
> > * Regarding your question, we admit that it is challenging to analyze the training of vanilla DEQ with GD except for some special cases such as in the NTK regime. Since this paper aim to analyze the bias of DEQ beyond NTK regime, we make simplifications on the model architecture to ensure tractability. Although it has some limitations, our model retains the implicit nature of DEQ and is essentially a nonlinear model. Thus, we believe our result provide some insights into the bias of standard DEQ (Please refer to Author Response and our response to Review HjzA for the weakness and Question 5 for more details). As for our focus on GD, it is worth noting that the primary optimization algorithms used for DEQ in real applications are also gradient-based algorithms such as variants of SGD and Adam (e.g., see [2,3] in our Reference). Since our paper proposes the first study on the bias of DEQ (beyond the NTK regime), we focus on the fundamental deterministic gradient algorithms. It is an interesting question to study the convergence and bias for usual DEQ under more general settings or with different optimization algorithms such as stochastic methods. We will include statements in our conclusion section and leave them as future works.
> >
> >
> > __Response to W3-2__:
> > According to Eq. (31)-(33) in Appendix A.3, the GF dynamic yields
> > $$\mathbf{\beta}(t)=\left(3\mathbf{X}^T \mathbf{v}(t)+\mathbf{\beta}(0)^{\odot -3}\right)^{\odot -\frac{1}{3}},$$
> > where $\mathbf{v}(t) = \int_{0}^t \left(\mathbf{X}\mathbf{\beta}(s) - \mathbf{y}\right) \mathrm{d}s$.
> > Thus it is impossible for any entry of $\mathbf{\beta}(t)$ to take value $0$ at any finite time $\tau$ since every entry $\mathbf{X}^T \mathbf{v}(\tau)$ cannot take infinity. Therefore, by using the continuity of $\mathbf{\beta}(t)$ and the fact that $\mathbf{\beta}(0)>0$, we know that $\mathbf{\beta}(t)$ remains positive throughout the training process.
> >
> > Thank you again for your valuable comments and questions. We hope our responses address your concerns. Please feel free to reach out if you have any further questions.

---

> > > ### Comment · Reviewer_j6dr · 2024-08-10
> > > **Thank you**
> > >
> > > I appreciate the authors for their kind and detailed responses. Now all my previous concerns are resolved.
> > >
> > > I asked an additional question regarding W3-2 because I got confused at some point in the proof. For now, let's just consider $d=1$ for simplicity.
> > > Previously, I thought that the deduction of Equation (32) from Equation (31) implicitly used the fact $\beta(t) \ne 0$ throughout the whole gradient flow dynamics, because the derivation requires **dividing** Equation (31) with $\beta(t)^{4}$.
> > > However, I realized that, if we assume $\beta(T) = 0$ at some time $T>0$, it yields a contradiction given all the assumptions in Theorem 3. Here is how it goes: suppose $T$ is the very first time that $\beta(T)=0$, i.e., $\beta(t)> 0$ for all $t\in [0, T)$. Clearly, Equation (33) is valid for all $t\in [0, T)$. So fix any $\epsilon \in (0, T)$. Then by Equation (33), we have $\beta(T-\epsilon) = (3X^{\top} v(T-\epsilon) + \beta(0)^{-3})^{-\tfrac13}$. By continuity of $\beta(t)$, it yields $0 = (3X^{\top} v(T) + \beta(0)^{-3})^{-\tfrac13}$, which can only happen when $X^{\top} v(T) = \infty$ (contradiction).
> > >
> > > I think this deduction is quite basic (with the background knowledge of ODE) but I missed it. Thus, I don't think the authors need to add more details to their paper like above. Thank you again for your kind explanation.
> > >
> > > I raise my score to 6.

---

> > > > ### Author Response · Authors · 2024-08-10
> > > > **Thank you**
> > > >
> > > > Thank you again for your valuable feedback. We will continue improving our paper.

---

### Official Review · Reviewer_HjzA · 2024-07-17

**Soundness:** 3
**Presentation:** 2
**Contribution:** 3
**Rating:** 5
**Confidence:** 4

**Summary:**

This paper explores the theoretical foundations of Deep Equilibrium Models (DEQ) and their advantages over Fully Connected Neural Networks (FNNs). It demonstrates DEQs have superior expressive power compared to FNNs of similar sizes, particularly in generating linear regions and approximating steep functions; and it also analyzes the implicit regularization and biases effects of DEQs in learning dynamics, which can lead to improved generalization, especially in out-of-distribution tasks. The paper supports these theoretical findings with empirical results.

**Strengths:**

The paper addresses a significant gap in the literature by providing a detailed theoretical analysis of the expressive power and learning dynamics of DEQs, which is a relatively new model in the field of machine learning. By providing a deeper understanding of DEQs' expressivity and learning dynamics, the paper opens new avenues for research in neural network architectures and their applications.

**Weaknesses:**

The assumption in Equation (9) that DEQs favor dense features might be misleading. The model only updates the diagonal in the weights while keeping the rest as zeros, effectively constraining the model to be sparse. It is straightforward to claim that the diagonals are dense since that is the only part of the model being utilized.

The paper misses citing some important related works, such as [1] which studies the convergence of linear DEQ, and [2] which explores information propagation in DEQ and shows its equivalence to a Gaussian process in the infinite-width limit.

[1] On the Theory of Implicit Deep Learning: Global Convergence with Implicit Layers, ICLR 2021
[2] Wide Neural Networks as Gaussian Processes: Lessons from Deep Equilibrium Models, NeurIPS 2023

**Questions:**

- DEQs differ from FNNs in three distinct ways: shared weights, input injection, and infinite depth. While the paper claims that DEQs are superior to FNNs in terms of expressivity due to their infinite depth, can the authors comment on or envision the significance of the other two features (shared weights and input injection)? How do these contribute to the overall performance and expressivity of DEQs?
- Comparing to FCN, actually DEQ has 3 distinct propoaretis, that is shared weights, input injection, and infinitely depth. The authors claim DEQ is better than FCN in terms of expressivity simply because of the infinitely depth. Can authors comments or envision the significance of the other 2 features?
- How do the authors envision the practical applications of their findings? Are there specific domains where DEQs could significantly outperform FNNs?
- Could the authors elaborate on the potential limitations of DEQs, especially in terms of computational complexity and training stability?
- How do the authors address the sparsity constraint in their model assumptions? Can they provide more justification for why the diagonals should be considered dense?
- Can the authors conduct experiments to show that the learned features are indeed dense? These experiments may include both the diagonal models in Equation (9) and the general models using the entire weight matrices $W$

**Limitations:**

See the weakness and questions.

---

> ### Author Rebuttal · Authors · 2024-08-07
>
> Thank you for your careful review and thoughtful questions and comments. Please see our response below.
>
> > The assumption in Equation (9) that DEQs favor dense features might be misleading. …. It is straightforward to claim that the diagonals are dense since that is the only part of the model being utilized.
>
> While our simplified model in Eq. (9) has some limitations to ensure solvability, we do not agree that our result is a direct consequence of the model assumption. In fact, updating only the diagonal entries does not necessarily lead to dense diagonals. As a counterexample, it is known that for matrix sensing or one-hidden-layer neural networks with quadratic activation (see Theorem 1.2 in [37] in our references for details), Gradient Descent will converge to a low-rank (diagonal sparse, not dense) solution. Specifically, consider the model $f_U(x) = \mathbf{1}^T q(U^Tx)$ with labels $y= \mathbf{1}^T q((U^*)^T x)$, where $q(\cdot)$ is the quadratic activation, $U$ is diagonal and the ground truth $U^*$ is sparse diagonal. It is shown that $\lVert UU^T-U^*(U^*)^T\rVert _F$ can be sufficiently small after some gradient steps by adjusting the initialization scale of $U$. Thus, the diagonal entries of $U$ are sparse even if they are the only part of the model being utilized. So we argue that the bias of DEQs in our paper is essentially due to the network architecture, rather than the model assumption in Eq. (9). Unlike FNNs that often have the so-called simplicity bias (prefer learning minimum norm or sparse solutions), DEQ tends to learn some hard problems. We will add more explanations in the revised version.
>
> > The paper misses citing some important related works.
>
> Thanks for your feedback. We will include the papers you mentioned in related works in our final version of the paper.
>
> > Can the authors comment on or envision the significance of the other two features (shared weights and input injection)? How do these contribute to the overall performance and expressivity of DEQs?
>
> Our separation results have leveraged all three distinct properties. Theorem 2 uses both the shared weights and infinite depth so that the feature of DEQ can be characterized as the solution to a fixed-point equation. And without the input injection, the fixed point is not even a function of the network input. As for the separate significance of shared weights and input injection on expressivity, the shared weights enhance the parameter efficiency of the model in approximating certain functions. The input injection can be regarded as a skip connection for the weight-untied model, potentially improving the expressive power, much like how skip connections in ResNet enhance its expressive power. These properties together enable DEQ to be efficient in approximating specific functions such as those as solutions to fixed-point equations or those with relatively large derivatives.
>
>
> > How do the authors envision the practical applications of their findings? Are there specific domains where DEQs could significantly outperform FNNs?
>
> A high-level conjecture from our analysis is that DEQ has potential advantages in learning certain high-frequency components. In Appendix B.2 in our paper, we conduct an experiment on audio representation. We observe that DEQ outperforms FNN with a noticeable error, preliminarily validating our findings and hypothesis. Based on our results, we envision that DEQs may outperform FNNs in the area of audio. We will discuss this in the revised version. However, we leave the full study as a future work.
>
> > Could the authors elaborate on the potential limitations of DEQs, especially in terms of computational complexity and training stability?
>
> The limitations of DEQs in terms of computational complexity and training stability are not the main consideration of this paper. Because the paper mainly studies the bias of DEQ caused by its architecture and learning dynamics.  However, we will add more review in the related work section for the limitations. From our understanding, one stability issue of training DEQ is ensuring the existence a unique fixed-point throughout training. The well-posedness may be violated for vanilla DEQ when $\lVert W\rVert _2$ is large, especially when trained using a large learning rate. Another stability issue is gradient explosion, similar to that in training RNNs. Specifically, the gradient w.r.t. $W$ involves $(I-W)^{-1}$. If $(I-W)$ is nearly singular, the gradient may explode.
>
> > How do the authors address the sparsity constraint in their model assumptions? Can they provide more justification for why the diagonals should be considered dense?
>
> Our model assumption in Eq. (9) is primarily for ensuring the tractability of the model due to the technical issues arising from the non-commutativity of matrices, especially as we aim to analyze the bias of DEQ beyond the NTK regime. This is a commonly considered tool that simply reduces a matrix problem to a vector problem (e.g., see work [29, 37], all these works start from analyzing vector problem and then consider matrix version). Again, we emphasize that diagonalization does not necessarily lead to dense diagonals. Therefore, we believe that this simplified model can still reveal the bias of general nonlinear DEQ to an extent. We hope this along with our response to the weakness above can address your concerns.
>
> > Can the authors conduct experiments to show that the learned features are indeed dense?
>
> Thanks for the suggestions. In the revised version, we add experiments to evaluate the feature density of a depth-$3$ FFN, a diagonal DEQ, and a vanilla DEQ for the GOTU task in Eq. (15). We generate heatmaps of the features of both DEQ and the feature before the last layer of FFN. It is shown that both the heatmap of diagonal DEQ and the heatmap of vanilla DEQ are lighter than that of FFN, indicating the features are indeed dense. Please see our attached PDF file in Author Response for details.

---

### Official Review · Reviewer_qjxV · 2024-07-22

**Soundness:** 2
**Presentation:** 2
**Contribution:** 2
**Rating:** 5
**Confidence:** 3

**Summary:**

This paper studies the expressivity and inductive bias of deep equilibrium models. First, the authors generate a separation result for expressivity between a fully connected deep model model and a deep equilibrium model, showing that if the depth of the fully connected model scales as $m^{\alpha}$ for width $m$ and $\alpha \in (0,1)$, that there are more linear regions in the DEQ model of width $m$ which can achieve the full set of $2^m$ linear regions. Second, the authors provide an example of a "steep" target function that depends on a single dimension of the inputs and approximates a step function. For this target function, the FNN would require weights with large $\infty$-norm (that scales exponentially in the dimension) while the DEQ can achieve error $\epsilon$ with weights that scale as $1/\epsilon$. Lastly, the authors provide a study of diagonal linear DEQs and characterize the type of min-norm solution obtained by running gradient flow on this model, a norm $q(\beta) \sim 1/\beta^2 + C \beta$ for constant $C$ which penalizes both small and large parameters. This model is also evaluated on a special OOD task.

**Strengths:**

This paper addresses an interesting architecture, deep equilibrium models, which have received less study than traditional feedforward architectures. The paper is therefore well motivated and timely. The analysis of the inductive bias (norm that is minimized) due to gradient flow training is quite interesting and novel to my knowledge.

**Weaknesses:**

Some aspects of the comparison between deep FFN and DEQs were a bit unclear. Specifically, for the approximation result, I was wondering whether the comparison between depth $L \leq m^\alpha$ FFNs and DEQs was a fair comparison. (See my questions below).
The experiments that the authors provide do show some situations where DEQs outperform FFNs. However, I think it would be more compelling theoretically if the experiments could be used to support the theoretical claims made in the paper (ie varying $d$ in the steep target function and showing infinity norm of FFNs increases but not the DEQs, etc).

**Questions:**

1. A DEQ could be viewed as the result of applying an infinite depth network with shared weights. If depth were to be unrestricted, then this hypothesis class should be contained within the set of all infinite depth FFNs. Is it fair to compare finite depth FFNs to DEQs? Would it perhaps be more interesting to compare the solutions identified by gradient flow in the FFN (where weights are not tied) to DEQ (where weights are tied)?
2. In Remark 1, it is stated that an FFN could in principle achieve $2^{ m L }$ linear regions.  Supposing this could be achieved, wouldn't we expect a width $m$ and depth $L > 1$ FFN to have *more* linear regions than a DEQ model? Do the authors' results provide evidence that suggests that $2^{mL}$ linear regions cannot be achieved?
3. Should line 255 be $\sum_i \beta_i^{-2}$ instead of $\sum_i \beta_i^{-1}$ ?

**Limitations:**

The authors mention briefly that they study a diagonal linear DEQ for tractability, but they could include more limitations in the conclusion/discussion sections. They could also point out that some of their results are about approximation of particularly chosen target functions.

---

> ### Author Rebuttal · Authors · 2024-08-07
>
> Thank you for your careful review and thoughtful questions and comments. Please see our response below.
>
> > I was wondering whether the comparison between depth $L\leq mα$ FFNs and DEQs was a fair comparison. (See my questions below).
>
> Please see our response for the question Q1 below.
>
> > However, I think it would be more compelling theoretically if the experiments could be used to support the theoretical claims made in the paper (ie varying d in the steep target function and showing infinity norm of FFNs increases but not the DEQs, etc).
>
> Thanks for your feedback. In our experiment in Figure 1 for approximating the steep function in Theorem 2, we vary the input dimension $d$ from $10$ to $20$ as we set $\delta = 2^{-10}$ and $\delta = 2^{-20}$. To further support the theoretical results of Theorem 2, we add experiments varying $d$ from $5$ to $20$ for the steep target function and showing the infinity norm of weights of both networks. Our results show that the infinity norm of FFNs increases as we apply larger $d$, while the infinity norm of DEQs remains lower than that of FFNs, consistent with Theorem 2. Please see our attached PDF in Author Response for details.
>
> > A DEQ could be viewed as the result of applying an infinite depth network with shared weights. If depth were to be unrestricted, then this hypothesis class should be contained within the set of all infinite depth FFNs. Is it fair to compare finite depth FFNs to DEQs?
>
> We think that our comparison of the expressivity of DEQ and finite-depth FFN is fair.
> One primary goal for the comparison is to investigate how the DEQ architecture influences the model capacity with the same number of parameters as FFN whose weights are not tied. This analysis provides intuitions for understanding the model bias of DEQ. Note that one conceptual understanding here is ``no free lunch``. Under the same number of parameters, we study for what kind of function, DEQ is provably better than standard FFN. This paper follows the common ways to characterize expressivity and separations. Our separation results are mainly stated from the actual size of the networks (Theorem 2 also considers the parameter magnitude). Specifically, Theorem 1 compares the expressive power of DEQ with $O(m^2+md)$ parameters with a class of FFN (i.e., $L\leq m^{\alpha})$ with sub-exponentially many parameters by providing a separation result showing that DEQ is much more *parameter-efficient* in approximating certain target function. We will add more discussions to emphasize the fairness and importance of our comparisons in the final version of the paper.
>
> > Would it perhaps be more interesting to compare the solutions identified by gradient flow in the FFN (where weights are not tied) to DEQ (where weights are tied)?
>
> We agree that it is a very interesting problem. However, analyzing the FFN optimized by gradient flow is beyond the scope of model expressivity to an extent. The solvability of multilayer FFN remains widely open except for some special cases such as in the lazy regime of NTK. In this regime, the model requires to be heavily over-parameterized and approximates just a linear model. The analysis would mainly concern traditional methods, such as studying the decay rate of kernels. However, in Section 5, we take a further step by studying the dynamic bias (trained by GF and GD) of a simplified DEQ in comparison to that for FNN. Here though the model is simplified, it is non-linear and actually beyond the NTK regime. We leave the study in the NTK regime as future works. So we do have some results for studying the dynamic bias (implicit regularization) for DEQ.
>
> > In Remark 1, it is stated that an FFN could in principle achieve $2^{mL}$ linear regions. Supposing this could be achieved, wouldn't we expect a width $m$ and depth $L>1$ FFN to have more linear regions than a DEQ model?
>
> Thanks for the valuable question. As stated in Remark 1, our comparison of the number of linear regions between FFN and DEQ is actually stated in terms of the number of neurons (the input neurons are not counted in). Note that a DEQ with $mL$ neurons has width $mL$, thus it can generate $2^{mL}$ linear regions according to our Proposition 1. So an FFN with width $m$ and depth $L>1$, i.e., $mL$ neurons, cannot generate more linear regions than that of DEQ with the same neurons. Therefore, we claim that DEQ is can potentially generate a larger number of linear regions compared to FFNs with the same number of neurons.
>
> > Do the authors' results provide evidence that suggests that $2^{mL}$ linear regions cannot be achieved?
>
> The Lemma 5 in our Appendix A.1 may provide some evidence. It shows that when the input dimension is $1$ and the width of each layer is $m$ for a depth-$L$ ReLU-FFN, the number of linear regions is $O(m^L)$, strictly smaller than $2^{mL}$. Through our further investigation, it is shown in Theorem 1 in [1] (See the statement in Page 4 in [1]) that when the input dimension $m_0=O(1)$ and the width of each layer is $m$ for a depth-$L$ ReLU-FFN, the asymptotic upper bound is of $O(m^{Lm_0})$, which is of smaller order in magnitude compared with $2^{mL}$. We will add a statement to include the evidence in the final version.
> ***
> [1] Bounding and Counting Linear Regions of Deep Neural Networks, ICML 2018.
>
> > Should line 255 be $\sum_{i}\beta_{i}^{−2}$ instead of $\sum_{i}\beta_{i}^{−1}$?
>
> We are sorry for the typo. It should be $\sum_{i=1}^d \beta_i^{-2}$ in line 225 (I think you refer to line 225 instead of 255). We will revise it in the final version.
>
> > The authors mention briefly that they study a diagonal linear DEQ for tractability, but they could include more limitations in the conclusion/discussion sections. They could also point out that some of their results are about approximation of particularly chosen target functions.
>
> Thanks for your suggestion. We will add a discussion about our limitations in the conclusion section in the final version.

---

> > ### Comment · Reviewer_qjxV · 2024-08-10
> >
> > I thank the reviewers for their answers to my question and the attached rebuttal experiment. The justification to investigate architectures at fixed parameter counts makes sense and also the polynomial scaling of linear regions with width for FNN and exponential scaling for DEQ improves my understanding of the work. I will thus increase my score.

---

> > > ### Author Response · Authors · 2024-08-11
> > > **Thank you**
> > >
> > > Thank you again for your valuable feedback! We will continue improving our paper.

---

### Author Rebuttal · Authors · 2024-08-07

We express our sincere gratitude to all reviewers for the valuable and constructive comments.  Many of the suggestions will be incorporated in the final version of the paper.

Some reviewers have raised questions regarding the fairness of our comparison between DEQ and FFN and the validity of our simplification of the diagonal linear DEQ. We emphasize that our comparison is fair in terms of the number of network parameters, which is essential for understanding the model capacity and the parameter efficiency for certain functions. Its analysis provides intuitions for understanding the model bias of DEQ. We study for what kind of function, DEQ is provably better than standard FFN. Even when considering the extra cost in the forward propagation of DEQ, our comparisons remain valid (see our response to Reviewer j6dr for W1).

Regarding our model simplification in understanding the implicit bias of the Gradient Descent algorithm, it is mainly for ensuring solvability because we aim to analyze the bias of DEQ beyond the lazy training regime.  Similar kinds of simplification have been commonly considered in studying two-layer neural networks (e.g., see [29, 37] in our References). Our obtained results are not a direct consequence of our model assumption. The linear diagonal model preserves the implicit nature of DEQ and is essentially a nonlinear model. Thus, our results reveal the bias of general nonlinear DEQ to an extent.

Additionally, in response to the concerns raised, we have conducted additional experiments and included the results in the attached PDF file. It includes:

* Review qjxV: (1) Evaluation of the norm of weights of DEQ and FFN for approximating the steep function with varying $d$; Fig. 1.
* Review HjzA: An experiment on evaluating the density and norm of features; Fig. 2.
* Review j6dr: Comparison of DEQ with various FFNs having similar FLOPs per iteration; Table 1.
* Review QS3W: A saliency map experiment for a variant of ResNet-DEQ trained on image classification tasks; Fig. 3.

We hope that these additional results will address the reviewers’ concerns and strengthen our paper. Please let us know if you have any further questions!

---

### Comment · Area_Chair_NgaA · 2024-08-08

Just a friendly reminder to the reviewers to acknowledge the author's rebuttal, so that the discussion period can be used efficiently.

---

### Decision · Program_Chairs · 2024-09-25

**Decision:**

Accept (poster)

**Comment:**

The paper uncovers a number of interesting properties of Deep Equilibrium Models. The authors all lean towards acceptance.